# Label-Retrieval-Augmented Diffusion Models for Learning from Noisy Labels

**Jian Chen**[1]    **Ruiyi Zhang**[2]    **Tong Yu**[2]    **Rohan Sharma**[1]
**Zhiqiang Xu**[3]    **Tong Sun**[2]    **Changyou Chen**[1]
[1]University at Buffalo    [2]Adobe Research    [3]MBZUAI
{jchen378,rohanjag,changyou}@buffalo.edu
{ruizhang,tyu,tsun}@adobe.com   zhiqiang.xu@mbzuai.ac.ae

## Abstract

Learning from noisy labels is a long-standing problem in machine learning for real applications. One of the main research lines focuses on learning a label corrector to purify potential noisy labels. However, these methods typically rely on strict assumptions and are limited to certain types of label noise. In this paper, we reformulate the label-noise problem from a generative-model perspective, *i.e.*, labels are generated by gradually refining an initial random guess. This new perspective immediately enables existing powerful diffusion models to seamlessly learn the stochastic generative process. Once the generative uncertainty is modeled, we can perform classification inference using maximum likelihood estimation of labels. To mitigate the impact of noisy labels, we propose **L**abel-**R**etrieval-**A**ugmented (LRA) diffusion model [1], which leverages neighbor consistency to effectively construct pseudo-clean labels for diffusion training. Our model is flexible and general, allowing easy incorporation of different types of conditional information, *e.g.*, use of pre-trained models, to further boost model performance. Extensive experiments are conducted for evaluation. Our model achieves new state-of-the-art (SOTA) results on all standard real-world benchmark datasets. Remarkably, by incorporating conditional information from the powerful CLIP model, our method can boost the current SOTA accuracy by 10-20 absolute points in many cases.

## 1 Introduction

Deep neural networks have achieved extraordinary accuracy on various classification tasks. These models are typically trained through supervised learning using large amounts of labeled data. However, large-scale data labeling could cost huge amount of time and effort, and is prone to errors caused by human mistakes or automatic labeling algorithms [1]. In addition, research has shown that the ability of deep neural network models to fit random labels can result in reduced generalization ability when learning with corrupted labels [2]. Therefore, robust learning methods using noisy labels are essential for applying deep learning models to cheap yet imperfect datasets for supervised learning tasks.

There are multiple types of label noise investigated by previous research. More recent research has focused on studying the more realistic feature-dependent label noise, where the probability of mislabeling a given instance depends on its characteristics. This type of noise is more consistent with label noise in real-world datasets [3, 4, 1, 5, 6, 7]. To address this type of noise, a model is expected to be able to estimate the uncertainty of each training label. Many state-of-the-art methods have primarily relied on the observation that deep neural networks tend to learn simple patterns before

---

[1]Code is available at https://github.com/puar-playground/LRA-diffusion

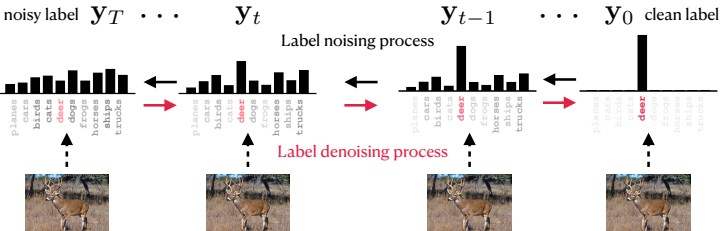

Figure 1: Label denoising as a reverse noising process.

memorizing random noise [8, 9]. This means a temporary phase exists in the learning process, where the model has learned useful features but has yet not started overfitting corrupt labels. At this stage, the model predictions can be used to modify the training labels, so that they are more consistent with model predictions [10, 11, 12, 13, 14, 15]. By correctly modifying the labels, the number of clean training sample increases, which can further benefit the training. These type of approaches, however, is inherently risky because the point at which the model starts to overfit varies with the network structure and dataset. Starting too early can corrupt the training data, while starting too late may not prevent overfitting [16]. Therefore, it is vital to carefully tune the hyper-parameters of the training strategy, such as the number of epochs for warm-up training, learning rate, and uncertainty threshold, to achieve successful training results.

Another class of methods adopts the assumption of label propagation in semi-supervised learning [17, 18], where nearby data points in a feature space tend to have the same label. Therefore, they use *neighbor consistency*[2] regularization to prevent overfitting of the model [19, 20]. The performance highly depends on the quality of the encoder that maps the data to the feature space, as retrieving a neighbor that belongs to a different class could further mislead the training process. Encoders are therefore required to first learn high-level features of the data that can be used for classification, which could be trained simultaneously with the classifier using noisy labels. However, the training can also lead to overfitting or underfitting.

In this paper, by contrast, we formulate the label noise problem from a generative-model perspective, which naturally provides new insights into approaching the problem. Our intuition is to view the noisy labeling process as a stochastic label generation process. Thus, we propose to adopt the powerful diffusion model as the generative building block. Figure 1 illustrates our intuition. In the generative process, we start with a noisy estimation of the label, then gradually refine it to recover the clean label, which is equivalent to the reverse denoising process of the diffusion model.

Specifically, the diffusion model takes a noisy label and some useful conditional information (to be specified) as inputs, and learns to recover/generate the ground-truth labels as outputs. One challenge in this setting is that only noisy labels are available in practice for training. To overcome this issue, we adopt the principle of *neighbor consistency*, and propose *label-retrieval augmentation* to construct pseudo clean labels for diffusion model training, where a pre-trained image encoder is used to define the neighborhood of a sample. It is worth noting that the pre-trained image encoder would not be affected by the label noise, because they can be trained in a self-supervised manner [21, 22] or on an additional clean dataset [23, 24]. In fact, pre-training can tremendously improve the model's adversarial robustness [25] and has been used to improve model robustness to label corruption [26]. Another merit of our design is that it is general enough to allow natural incorporation of powerful large pre-trained model such as the CLIP model to further boost the performance.

In addition, the probability nature of diffusion models can also be better equipped to handle uncertainty in the data and label, thus providing more robust and accurate predictions. We call our model LRA-diffusion (label-retrieval-augmented diffusion).

Our main contributions are summarized as follows:

- We formulate learning from noisy labels as modeling a stochastic process of conditional label generation, and propose to adopt the powerful diffusion model to learn the conditional label distribution.

---

[2]Nearby data points tend to have the same label.

- We incorporate the neighbor consistency principle into the modeling, and design a novel label-retrieval-augmented diffusion model to learn effectively from noisy label data.

- We further improve our model by incorporating auxiliary conditional information from large pre-trained models such as CLIP.

- Our model achieves the new state-of-the-art (SOTA) in various real-world noisy label benchmarks, *e.g.*, 20% accuracy improvement on noisy CIFAR-100 benchmark.

## 2 Preliminary

Diffusion models were initially designed for generative modeling. Recently, it has been extended for classification and regression problems. In this section, we introduce the Classification and Regression Diffusion Models (CARD) [27], which our model is based on.

The CARD model transforms deterministic classification into a conditional label generation process, allowing for more flexible uncertainty modeling in the labeling process [27]. Similar to the standard diffusion model, CARD consists of a forward process and a reverse process. In the forward process, an $n$-dimensional one-hot label $\mathbf{y}_0$ is gradually corrupted to a series of intermediate random vectors $\mathbf{y}_{1:T}$, which converges to a random variable with a multi-variant Gaussian distribution $\mathcal{N}(f_q(\mathbf{x}), \mathbf{I})$ (latent distribution) after $T$ steps, where the mean is defined by a pre-trained $n$-dimensional image encoder $f_q$. The transition steps between adjacent intermediate predictions is modeled as Gaussian distributions, $q(\mathbf{y}_t|\mathbf{y}_{t-1}, f_q) = \mathcal{N}(\mathbf{y}_t; \boldsymbol{\mu}_t, \beta_t \mathbf{I})$, with mean values $\boldsymbol{\mu}_1, \cdots, \boldsymbol{\mu}_T$ and a variance schedule $\beta_1, \cdots, \beta_T$, where $\boldsymbol{\mu}_t = \sqrt{1 - \beta_t}\mathbf{y}_{t-1} + (1 - \sqrt{1 - \beta_t})f_q(\mathbf{x})$. This admits a closed-form sampling distribution, $q(\mathbf{y}_t|\mathbf{y}_0, f_q) = \mathcal{N}(\mathbf{y}_t; \boldsymbol{\mu}_t, (1 - \bar{\alpha}_t)\mathbf{I})$, with an arbitrary timestep $t$ and $\boldsymbol{\mu}_t = \sqrt{\bar{\alpha}_t}\mathbf{y}_0 + (1 - \sqrt{\bar{\alpha}_t})f_q(\mathbf{x})$. The mean term can be viewed as an interpolation between true data $\mathbf{y}_0$ and the mean of the latent distribution $f_q(\mathbf{x})$ with a weighting term $\bar{\alpha}_t = \prod_t (1 - \beta_t)$.

In the reverse (generative) process, CARD reconstructs a label vector $\mathbf{y}_0$ from an $n$-dimensional Gaussian noise $y_T \sim \mathcal{N}(f_q(\mathbf{x}), \mathbf{I})$ by approximating the denoising transition steps conditioned on the data point $\mathbf{x}$ and another pre-trained image encoder $f_p$ in an arbitrary dimension. The transition step is also Gaussian for an infinitesimal variance $\beta_t$ [28] (define $\tilde{\beta}_t = \frac{1 - \bar{\alpha}_{t-1}}{1 - \bar{\alpha}_t}\beta_t$):

$$p_\theta(\mathbf{y}_{t-1}|\mathbf{y}_t, \mathbf{x}, f_p) = \mathcal{N}(\mathbf{y}_{t-1}; \boldsymbol{\mu}_\theta(\mathbf{y}_t, \mathbf{x}, f_p, t), \tilde{\beta}_t \mathbf{I}). \tag{1}$$

The diffusion model is learned by optimizing the evidence lower bound with stochastic gradient descent:

$$\mathcal{L}_{\text{ELBO}} = \mathbb{E}_q \left[ \mathcal{L}_T + \sum_{t>1}^{T} \mathcal{L}_{t-1} + \mathcal{L}_0 \right], \text{ where} \tag{2}$$

$\mathcal{L}_0 = -\log p_\theta(\mathbf{y}_0|\mathbf{y}_1, \mathbf{x}, f_p)$, $\mathcal{L}_{t-1} = D_{\text{KL}}(q(\mathbf{y}_{t-1}|\mathbf{y}_t, \mathbf{y}_0, \mathbf{x}, f_q)||p_\theta(\mathbf{y}_{t-1}|\mathbf{y}_t, \mathbf{x}, f_p))$
$\mathcal{L}_T = D_{\text{KL}}(q(\mathbf{y}_T|\mathbf{y}_0, \mathbf{x}, f_q)||p(\mathbf{y}_T|\mathbf{x}, f_p))$.

Following [29], the mean term is written as $\boldsymbol{\mu}_\theta(\mathbf{y}_t, \mathbf{x}, f_p, t) = \frac{1}{\sqrt{\alpha_t}}(\mathbf{x}_t - \frac{\beta_t}{\sqrt{1-\bar{\alpha}_t}}\boldsymbol{\epsilon}_\theta(\mathbf{y}_t, \mathbf{x}, f_p, t))$ and the objective can be simplified to $\mathcal{L}_{\text{simple}} = \|\boldsymbol{\epsilon} - \boldsymbol{\epsilon}_\theta(\mathbf{y}_t, \mathbf{x}, f_p, t)\|^2$.

## 3 Label-Retrieval-Augmented Diffusion Model

Inspired by CARD, Label-Retrieval-Augmented (LRA) diffusion models reframe learning from noisy labels as a stochastic process of conditional label generation (*i.e.*, label diffusion) process. In this section, we first provide an overview of the our model in Section 3.1 and then introduce the proposed label-retrieval-augmented component in Section 3.2, which can leverage label consistency in the training data. Next, we introduce an accelerated label diffusion process to significantly reduce classification model inference time in Section 3.3. Finally, a new conditional mechanism is proposed to enable the usage of pre-trained models in Section 3.4.

### 3.1 Model Overview

Our overall label-retrieval-augmented diffusion model is illustrated in Figure 2, where a diffusion model is adopted for progressively label denoising, by leveraging both the retrieved labels and auxiliary information from pre-trained models. Our model employs two pre-trained networks, denoted as $f_q$ and $f_p$ encoders, to encode conditional information that facilitates the generation process. The $f_q$ encoder serves as a mean estimator for $\mathbf{y}_T$, providing an initial label guess for a given image. This encoder could be a standard classifier trained on noisy labels. On the other hand, the $f_p$ encoder operates as a high-dimensional feature extractor, assisting in guiding the reverse procedure. $\mathbf{y}_t$ and $f_p(\mathbf{x})$ are concatenated together before being processed. Details about our neural-network architecture design are provided in Supplementary C.

During training, we use labels retrieved from the neighborhood as the generation target $\mathbf{y}_0$. Then, in the forward process, the distribution of neighboring labels is progressively corrupted towards a standard Gaussian distribution centered at the estimated mean $f_q(\mathbf{x})$. During testing, we employ a generalized DDIM method to efficiently compute the maximum likelihood estimation of $\mathbf{y}_0$.

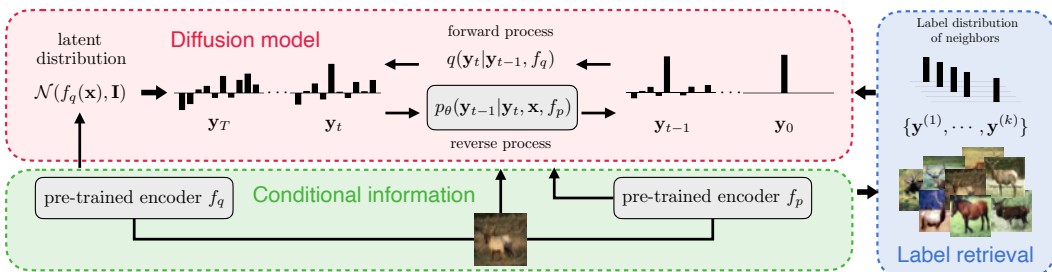

Figure 2: Overview of the proposed framework for improving learning performance from noisy labels. The figure depicts the three main components, including (1) using diffusion models to imitate and inverse the label noising process; (2) using pre-trained encoders (*i.e.*, $f_q$ and $f_p$) within the diffusion model, and (3) the label-retrieval-augmentation approach using the $f_p$ encoder to encourage neighbor consistency of image labels.

### 3.2 Label-retrieval Augmentation for Training

The noisy nature of labels excludes the availability of clean labels for training. To mitigate the issue, we propose a training strategy based on the concept of retrieval augmented learning [30, 31], such that it is more resistant to label noise. Our main assumption is that in a latent space, data points from different classes form distinctive clusters. Therefore, the majority of a data point's neighbors are expected to have the same label as the point itself. To this end, we used a pre-trained encoder, illustrated as $f_p$ in Figure 2, to map the data into an embedding space and retrieve a label $y'$ from labels of the $k$ nearest neighbors $\{y^{(1)}, \cdots, y^{(k)}\}$ in the training set. The diffusion model was then trained to learn the conditional distribution $p(y'|\mathbf{x})$ of labels within the neighborhood, rather than the distribution of labels $p(y|\mathbf{x})$ for the data point itself. We empirically select the value of $k$ based on the KNN accuracy obtained on the validation data, as detailed in Supplementary Section C.

Label-retrieval augmentation enables the model to make use of the information from multiple and potentially more accurate labels to improve its prediction performance. Algorithm 1 describes the training procedure. Additionally, diffusion models are known to be effective at modeling multimodal distributions. By training the model to generate different labels from neighbors based on the same data point, the model can produce stochastic predictions based on the distribution to capture the uncertainty inherent in the data labeling process. As a result, the trained model can be used not only as a classifier, but also as a sampler that simulates the actual labeling process.

### 3.3 Efficient Inference with generalized DDIM

The iterative generation nature of the classification diffusion model makes its inference efficiency not comparable to traditional classifiers. To enhance the inference efficiency, we propose to incorporate the efficient sampling methods, Denoising Diffusion Implicit Model (DDIM) [32], to accelerate the

---

**Algorithm 1** Training

---

**Input:** training set $\{\mathbf{X}, \mathbf{Y}\}$, image encoder $f_p$, $f_q$.

1: **while** not converged **do**
2:     Sample data $(\mathbf{x}, y) \sim \{\mathbf{X}, \mathbf{Y}\}$; time slice $t \sim \{1, \cdots, T\}$; and noise $\boldsymbol{\epsilon} \sim \mathcal{N}(0, \mathbf{I})$
3:     Retrieve labels $\{y^{(1)}, \cdots, y^{(k)}\}$ of neighbors in the feature space defined by the encoder $f_p$
4:     Sample $y' \sim \{y, y^{(1)}, \cdots, y^{(k)}\}$, and convert it to a one-hot vector $\mathbf{y}_0$
5:     Take gradient descent step on the loss:

$$\left\| \boldsymbol{\epsilon} - \boldsymbol{\epsilon}_\theta(\sqrt{\bar{\alpha}_t}\mathbf{y}_0 + (1 - \sqrt{\bar{\alpha}_t})f_q(\mathbf{x}) + \sqrt{1 - \bar{\alpha}_t}\boldsymbol{\epsilon}, \mathbf{x}, f_p(\mathbf{x}), t) \right\|^2$$

6: **end while**

---

label diffusion process. However, the utilization of the mean estimator $f_q$ makes DDIM incompatible with our setting, as our generation process begins with a non-zero mean Gaussian distribution $\mathcal{N}(f_q(\mathbf{x}), \mathbf{I})$. Therefore, we adjust the DDIM method into a more general form that fits our framework. Analogous to DDIM, our sampling process maintains the same marginal distribution as the original $q(\mathbf{y}_t|\mathbf{y}_0, f_q)$ closed-form sampling process. Detailed derivations can be found in Supplementary A.

With DDIM, the trained model can generate a label vector in much less steps following a pre-defined sampling trajectory of $\{T = \tau_S >, \cdots, > \tau_1 = 1\}$, where $S < T$. Consequently, $\mathbf{y}_t$ can be computed as:

$$\mathbf{y}_{\tau_s} = \sqrt{\bar{\alpha}_{\tau_s}}\mathbf{y}_0 + (1 - \sqrt{\bar{\alpha}_{\tau_s}})f_q(\mathbf{x}) + \sqrt{1 - \bar{\alpha}_{\tau_s}}\boldsymbol{\epsilon}, \tag{3}$$

where $\boldsymbol{\epsilon} \sim \mathcal{N}(\mathbf{0}, \mathbf{I})$. Similar to CARD [27], we predict the *denoised label* $\tilde{\mathbf{y}}_0$, a prediction of $\mathbf{y}_0$ given $\mathbf{y}_{\tau_s}$, as:

$$\tilde{\mathbf{y}}_0 = \frac{1}{\sqrt{\bar{\alpha}_{\tau_s}}}[\mathbf{y}_{\tau_s} - (1 - \sqrt{\bar{\alpha}_{\tau_s}})f_q(\mathbf{x}) - \sqrt{1 - \bar{\alpha}_{\tau_s}}\boldsymbol{\epsilon}_\theta(\mathbf{y}_{\tau_s}, \mathbf{x}, f_p(\mathbf{x}), \tau_s)]. \tag{4}$$

When $\tau_{s-1} > 0$, we can compute $\mathbf{y}_{\tau_{s-1}}$ given $\mathbf{y}_{\tau_s}$ from the non-Markovian forward process defined as:

$$\mathbf{y}_{\tau_{s-1}} = \sqrt{\bar{\alpha}_{\tau_{s-1}}}\tilde{\mathbf{y}}_0 + (1 - \sqrt{\bar{\alpha}_{\tau_{s-1}}})f_q(\mathbf{x}) + \sqrt{1 - \bar{\alpha}_{\tau_{s-1}}} \cdot \boldsymbol{\epsilon}_\theta(\mathbf{y}_{\tau_s}, \mathbf{x}, f_p(\mathbf{x}), \tau_s). \tag{5}$$

As the dimension of label vectors is usually much lower than that of an image, the model can employ fewer steps in the reverse process without compromising generative quality. In our experiments, we use $S = 10$ and $T = 1000$, substantially reducing the time cost of the classification process. Supplementary Figure B gives an example of the label generation (classification) process on the CIFAR-10 dataset.

To further enhance the inference efficiency, we propose a simple and effective trick for computing the maximum likelihood estimation of labels. As the generative process is deterministic given $\mathbf{y}_T$, which is sampled from a uni-modal Gaussian distribution, we approximate the maximum likelihood estimation of labels by initiating from the mean, *i.e.*, $\mathbf{y}_0 = \text{DDIM}(\mathbf{y}_T = f_q(\mathbf{x}), \mathbf{x})$. This trick circumvents the time-consuming majority voting approximation that demands repeated generation.

### 3.4 Flexible conditioning with pre-trained encoders

The original CARD model employs a single model for both the $f_p$ and $f_q$ encoders. However, this limits their representation capacity [33] as the dimension of $f_q(\mathbf{x})$ is typical relatively small, *i.e.*, equalling the number of classes. To mitigate this and improve model performance, we abandon the assumption that $f_p = f_q$, enabling the use of a more powerful pre-trained encoder (*e.g.*, the CLIP image encoder [24]) with arbitrary dimensions for $f_p$.

Empirically, we find that the model can still achieve satisfactory performance when the magnitude of $f_q(\mathbf{x})$ is small, which means the latent representation $\mathbf{y}_T = f_q(\mathbf{x}) + \boldsymbol{\epsilon}$ is dominated by the noise term $\boldsymbol{\epsilon} \sim \mathcal{N}(\mathbf{0}, \mathbf{I})$. In this case, the information provided by $f_q(\mathbf{x})$ to the diffusion process is limited. As a result, we simply set $f_q(\mathbf{x}) = \mathbf{0}$ to avoid handling an additional n-dimensional $f_q$ encoder. For the $f_p$ encoder, one can employ flexible pre-trained models as presented in Section 5. In this paper, we use the SimCLR model trained on the training images (without supervised information) and the pre-trained CLIP model.

# 4    Related work

**Robust loss function and regularization techniques.**    Several noise-robust loss functions and regularization techniques have been proposed as alternatives to the commonly used cross-entropy loss (CE), which is not robust to label noise. Mean absolute error (MAE) [34] loss has been shown to be robust against noisy labels. Generalized Cross-Entropy (GCE) [35] combines CE and MAE for faster convergence and better accuracy. Symmetric cross-entropy Learning (SL) [36] couples CE with a noise-robust counterpart and has been found to have higher performance than GCE, particularly for high noise rates. Label Smoothing Regularization [37] alleviates overfitting by linearly combining labels with a uniform distribution. Bootstrapping technique [38] combines the labels with the current model prediction. Dynamic bootstrapping [39, 40] uses the prediction confidence to control the weighting in the combination. Neighbor Consistency Regularization (NCR) [19] encourages consistency of prediction based on learned similarity. Our method is also based on the principle of neighbor consistency. However, instead of encouraging consistent predictions among neighbors, our model directly learns from the labels of neighbors. This allows for estimating instance-level uncertainty by learning the label distribution among neighbors, rather than learning a point estimation.

**Data recalibration.**    Data recalibration techniques progressively remove or correct mislabeled data during training to improve the reliability of training data. Wang et al. [11] used the learned similarity and label consistency to identify and discard data with noisy labels. TopoFilter [41] selects clean data by analyzing the topological structures of the training data in the learned feature space. Cheng et al. [4] defines a Bayes optimal classifier to correcct labels. Zheng et al. [14] proposed using a likelihood ratio test (LRT) to correct training labels based on predictions. Zhang et al. [15] used LRT to correct labels progressively and provides a theoretical proof for convergence to the Bayes optimal classifier. Dividemix [42], LongReMix [43], and CC [44] treat the low confident data as unlabeled, and then employ semi-supervised learning algorithms [45] for further analysis. C2D [46] combines Dividemix with self-supervised pre-training to boost its performance by improving the quality of the extracted features. Our approach employs the same assumption as TopoFilter that data belonging to the same class should be clustered together with ideal feature representations. However, our technique isn't confined to learned features potentially distorted by label noises. Instead, similar to C2D, our method can effectively leverage the high-quality feature learned by pre-trained encoders to achieve superior accuracy.

**Guided diffusion model and retrieval augmentation.**    Guided diffusion is a technique applied to diffusion models for conditional generation. Classifier guidance [47] is an cost-effective method leveraging the gradient of a classifier to steer the generative process of a trained diffusion model. On the other hand, Classifier-free guidance [48] learns the conditional distribution during training for improved generation quality. This approach also allows for the use of continuous guidance information, such as embedding vectors, rather than being limited to discrete labels. Classification and Regression Diffusion Models (CARD) [27] formulates classification and regression as a conditional generation task that generates labels or target variables conditioned on images. Our approach follows the same paradigm, and leverages the multi-modal coverage ability of diffusion models to learn the label distribution within the neighborhood. SS-DDPM [49] proposes a diffusion model with arbitrary noising distributions defined on constrained manifold, e.g., the probabilistic simplex for label generation. Retrieval-augmented diffusion models [30] used retrieved neighbors from an external database as conditional information to train diffusion models for image synthesis. Retrieval Augmented Classification [31] used retrieval-augmentation to train classification model using class-imbalanced training data. Our approach differs from theirs by retrieving labels instead of data to reduce label noise in training rather than increasing the training data. In addition, our model does not require an external database.

# 5    Experiments

We first evaluate the performance of our method on datasets with various types synthetic noises. Then, we perform experiments on four real-world datasets. To better understand the performance gain sources, we conduct ablation studies to measure the impacts of conditional diffusion and different pseudo-label construction strategies. All experiments were done on four NVIDIA Titan V GPUs. Comprehensive implementation details and hyper-parameters are provided in the Supplementary C.

Table 1: Classification accuracy (%) on CIFAR-10 and CIFAR-100 datasets under PMD noises and hybrid noises, combining PMD noise with Uniform (U) and Asymmetric (A) noise.

| Methods | CIFAR-10 | | | | |
| --- | --- | --- | --- | --- | --- |
| | 35% PMD | 70% PMD | 35% PMD + 30% U | 35% PMD + 60% U | 35% PMD + 30% A |
| Standard | 78.11 ± 0.74 | 41.98 ± 1.96 | 75.26 ± 0.32 | 64.25 ± 0.78 | 75.21 ± 0.64 |
| Co-teaching+ [50] | 79.97 ± 0.15 | 40.69 ± 1.99 | 78.72 ± 0.53 | 55.49 ± 2.11 | 75.43 ± 2.96 |
| GCE [35] | 80.65 ± 0.39 | 36.52 ± 1.62 | 78.08 ± 0.66 | 67.43 ± 1.43 | 76.91 ± 0.56 |
| SL [36] | 79.76 ± 0.72 | 36.29 ± 0.66 | 77.79 ± 0.46 | 67.63 ± 1.36 | 77.14 ± 0.70 |
| LRT [14] | 80.98 ± 0.80 | 41.52 ± 4.53 | 75.97 ± 0.27 | 59.22 ± 0.74 | 76.96 ± 0.45 |
| CC [44] | 81.23 ± 0.78 | 42.43 ± 1.56 | 79.6 ± 0.44 | 70.71 ± 0.34 | 78.66 ± 0.66 |
| PLC [15] | 82.80 ± 0.27 | 42.74 ± 2.14 | 79.04 ± 0.50 | 72.21 ± 2.92 | 78.31 ± 0.41 |
| SimCLR KNN | 83.71 | 29.45 | 78.25 | 54.82 | 75.37 |
| C2D + SimCLR [46] | 83.84 ± 0.13 | 34.23 ± 0.45 | 85.61 ± 0.29 | 81.39 ± 0.68 | 83.06 ± 0.57 |
| LRA-diffusion (SimCLR) | 88.76 ± 0.24 | 42.63 ± 1.97 | 88.41 ± 0.37 | 84.43 ± 0.82 | 85.64 ± 0.23 |
| CLIP KNN | 91.80 | 30.66 | 84.67 | 57.03 | 81.76 |
| LRA-diffusion (CLIP) | 96.54 ± 0.13 | 44.62 ± 0.18 | 95.71 ± 0.17 | 87.21 ± 0.71 | 93.65 ± 0.40 |

| Methods | CIFAR-100 | | | | |
| --- | --- | --- | --- | --- | --- |
| | 35% PMD | 70% PMD | 35% PMD + 30% U | 35% PMD + 60% U | 35% PMD + 30% A |
| Standard | 57.68 ± 0.29 | 39.32 ± 0.43 | 48.86 ± 0.56 | 35.97 ± 1.12 | 45.85 ± 0.93 |
| Co-teaching+ | 56.70 ± 0.71 | 39.53 ± 0.28 | 52.33 ± 0.64 | 27.17 ± 1.66 | 51.21 ± 0.31 |
| GCE | 58.37 ± 0.18 | 40.01 ± 0.71 | 52.90 ± 0.53 | 38.62 ± 1.65 | 52.69 ± 1.14 |
| SL | 55.20 ± 0.33 | 40.02 ± 0.85 | 51.34 ± 0.64 | 37.57 ± 0.43 | 50.18 ± 0.97 |
| LRT | 56.74 ± 0.34 | 45.29 ± 0.43 | 45.66 ± 1.60 | 23.37 ± 0.72 | 52.04 ± 0.15 |
| CC | 59.44 ± 0.33 | 42.79 ± 1.21 | 56.58 ± 0.45 | 43.64 ± 1.71 | 54.45 ± 1.22 |
| PLC | 60.01 ± 0.43 | 45.92 ± 0.61 | 60.09 ± 0.15 | 51.68 ± 0.10 | 56.40 ± 0.34 |
| SimCLR KNN | 54.22 | 39.25 | 51.87 | 41.73 | 46.50 |
| C2D + SimCLR | 69.28 ± 0.31 | 51.63 ± 0.53 | 59.87 ± 0.66 | 64.45 ± 1.41 | 65.65 ± 0.93 |
| LRA-diffusion (SimCLR) | 61.39 ± 0.15 | 53.37 ± 0.81 | 60.52 ± 0.28 | 55.79 ± 0.31 | 59.28 ± 0.11 |
| CLIP KNN | 79.58 | 52.55 | 69.66 | 50.91 | 61.19 |
| LRA-diffusion (CLIP) | 81.91 ± 0.10 | 74.52 ± 0.12 | 82.80 ± 0.11 | 81.10 ± 0.09 | 81.78 ± 0.15 |

## 5.1 Results on Synthetic Noisy Datasets

We conduct simulation experiments on the CIFAR-10 and CIFAR-100 datasets [51] to evaluate our method's performance under various noise types. Specifically, following [15], we test with *polynomial margin diminishing* (PMD) noise, a novel instance-dependent noise, at two noise levels and three hybrid noise types by adding *independent and identically distributed (i.i.d)* noises on top of instance-dependent noise.

For instance-dependent noise, we adopt the recently proposed *polynomial margin diminishing* (PMD) noise [15]. Following the original paper, we train a classifier $\boldsymbol{\eta}(x)$ using clean labels to approximate the probability mass function of the posterior distribution $p(y|\mathbf{x})$. Images are initially labeled as their most likely class $u_{\mathbf{x}}$ according to the predictions of $\boldsymbol{\eta}(x)$. Then, we randomly alter the labels to the second most likely class $s_{\mathbf{x}}$ for each image with probability: $p_{u_{\mathbf{x}},s_{\mathbf{x}}} = -\frac{c}{2}\left[\boldsymbol{\eta}_{u_{\mathbf{x}}}(\mathbf{x}) - \boldsymbol{\eta}_{s_{\mathbf{x}}}(\mathbf{x})\right]^2 + \frac{c}{2}$, where $c$ is a constant noise factor that controls the final percentage of noisy labels. Since corrupting labels to the second most likely class can confuse the "clean" classifier the most, it is expected to have the most negative impact on the performance of models learned with noisy labels. For PMD noise, we simulate two noise levels where 35% and 70% of the labels are corrupted.

For i.i.d noise, following [52, 15], we use a transition probability matrix $\mathbf{T}$ to generate noisy labels. Specifically, we corrupt the label of the $i$-th class to the $j$-th class with probability $T_{ij}$. We adopt two types of i.i.d noise in this study: (1) Uniform noise, where samples are incorrectly labeled as one of the other $(n-1)$ classes with a uniform probability $T_{ij} = \tau/(n-1)$ and $T_{ii} = 1 - \tau$, with $\tau$ the pre-defined noise level; (2) Asymmetric noise: we carefully design the transition probability matrix such that for each class $i$, the label can only be mislabeled as one specific class $j$ or remain unchanged with probability $T_{ij} = \tau$ and $T_{ii} = 1 - \tau$. In our experiment, we generated three types of hybrid noise by adding 30%, 60% uniform, and 30% asymmetric noise on top of 35% PMD noise.

We test our proposed label-retrieval-augmented diffusion model using two pre-trained encoders: (1) SimCLR [21]: We trained two encoders using the ResNet50 [53] architecture on the CIFAR-10 and CIFAR-100 datasets through contrastive learning; (2) CLIP [24]: the model is pre-trained on a large dataset comprising 400 million image-text pairs. Specifically, we used the vision transformer [54] encoder (ViT-L/14) with pre-trained weights, the best-performing architecture in CLIP. For

simplification, we refer to these configurations as LRA-diffusion (SimCLR) and LRA-diffusion (CLIP). We also investigated the performance of the KNN algorithm within the feature space defined by the SimCLR and CLIP encoders, denoted as SimCLR KNN and CLIP KNN respectively.

Table 1 lists the performance of the *Standard* method (train classifier using noisy labels), our method, and baseline methods for learning from noisy labels. The results in white rows are borrowed directly from [15]. We can see that using the SimCLR encoder in the LRA-diffusion method results in superior test accuracy on both CIFAR-10 and CIFAR-100 datasets compared to other baselines, without the need for additional training data. This is because the SimCLR encoder is trained in an unsupervised manner, making it immune to label noise, and it can effectively extract categorical features for accurate image retrieval. Therefore, when the correct labels dominate the label distribution in the neighborhood, training with the labels of the retrieved neighbor images allows the model to learn with more correct labels. C2D [46] also utilizes a pre-trained SimCLR encoder for initialization, but label noise may still affect the feature space during training. In contrast, our method freezes the feature encoder, shielding the pre-trained features from noise. Results on CIFAR-10 demonstrate that when the pre-trained feature has high KNN accuracy, our method performs better. On CIFAR-100, where the SimCLR feature has lower KNN accuracy, C2D is more effective, as it can refine the feature space through training. Moreover, freezing the feature encoder allows for efficient integration of large pre-trained encoders like CLIP, saving us from the prohibitive computational cost of fine-tuning. Notably, incorporating the CLIP encoder into our method significantly improves test accuracy over our LRA-diffusion (SimCLR) due to its excellent representation capabilities. In fact, by performing KNN in the CLIP feature space alone was able to achieve accuracy surpassing all competing methods in most experiments. This allows for the use of more clean labels during training, thus result in even higher accuracy.

## 5.2 Ablation Studies

To evaluate the contribution of diffusion and the pre-trained features, we conducted ablation experiments using CARD [27], SS-DDPM [49], and linear probing to incorporate pre-trained models. It is worth noting that the SimCLR model was trained on the same training set without access to external data. Results are given in Table 2.

Table 2: Classification accuracy (%) on CIFAR-10 and CIFAR-100 datasets with PMD noises using different combinations of model, pre-trained feature, and label.

| Methods | Feature space | Label | CIFAR-10 | | CIFAR-100 | |
|---|---|---|---|---|---|---|
| | | | 35% PMD | 70% PMD | 35% PMD | 70% PMD |
| Linear probing | SimCLR | noisy | 86.9 | 38.93 | 56.18 | 51.87 |
| Linear probing | SimCLR | sample | 63.8 | 35.84 | 53.34 | 52 |
| Linear probing | SimCLR | mean | 86.27 | 39.94 | 55.95 | 52.61 |
| ResNet+Linear | SimCLR | sample | 86.55 | 38.06 | 56.57 | 51.36 |
| CARD | SimCLR | sample | 75.08 | 34.35 | 52.03 | 32.67 |
| SS-DDPM (Dirichlet) | SimCLR | sample | 88.13 | 40.11 | 59.3 | 51.67 |
| LRA-diffusion (ours) | SimCLR | sample | **88.96** | **42.63** | **61.38** | **53.57** |
| Linear probing | CLIP | noisy | 85.35 | 37.4 | 65.02 | 53.21 |
| Linear probing | CLIP | sample | 95.61 | 40.17 | 63.98 | 58.53 |
| Linear probing | CLIP | mean | 96.19 | 35.19 | 69.05 | 62.76 |
| ResNet+Linear | CLIP | sample | 88.72 | 43.26 | 59.78 | 51.47 |
| CARD | CLIP | sample | 79.72 | 33.57 | 47.1 | 23.45 |
| SS-DDPM (Dirichlet) | CLIP | sample | 96.03 | 42.03 | 80.72 | 72.24 |
| LRA-diffusion (ours) | CLIP | sample | **96.55** | **44.51** | **81.92** | **74.58** |

Linear probing with sampled labels yielded lower accuracy than using noisy labels or the mean of neighboring labels. This difference may be due to the linear layer's inability to yield stochastic outputs from a multimodal distribution. During training, conflicting gradient directions may arise if the model tries to predict different labels across gradient steps, which can impede learning. However, due to the mode coverage ability of the diffusion model, our method can effectively learn from retrieval-augmented labels to generate different labels with different probabilities. We also test a baseline using an additional ResNet encoder along with the linear layer to mimic our model architecture shown in Figure C.1. The results are comparable with linear probing with sampled labels. Our model

significantly outperforms CARD, mainly due to the more informative $f_p$ encoder. We use the same model architecture for SS-DDPM, which uses Dirichlet distributions for noisy states. SS-DDPM performed slightly worse than our method, indicating that constraining noisy states within probability simplex may not benefit our task.

Additional ablation studies are included in the Supplementary D. The results demonstrate that in the absence of a pre-trained encoder, our model can leverage the features of a noisy classifier to enhance its accuracy. We also include results of ablation experiments using different approaches for incorporating the CLIP model. The results indicate that the superior performance of our method is not solely attributable to the strength of the CLIP features. In conclusion, our LRA-diffusion model provides an efficient approach for incorporating pre-trained encoders for learning from noisy labels.

## 5.3 Results on Real-world Noisy Datasets

We further evaluate the performance of our proposed method on real-world label noise. Following previous work [42, 15, 44, 43], we conducted experiments on four image datasets, *i.e.*, WebVision [55], ImageNet ILSVRC12 [56], Food-101N [57], and Clothing1M [58]. For experiments on Webvision, ILSVRC12, and Food-101N datasets, we use the CLIP image encoder as the $f_p$ encoder to train LRA-diffusion models. Comprehensive dataset description and implementation details can be found in the Supplementary C. We evaluated the performance of our method against a group of state-of-the-art (SOTA) methods. The results are presented in Table 3 and Table 4. Our approach significantly outperforms all the previous methods in terms of classification accuracy. It is important to highlight that EPL [59] incorporates the most powerful CLIP and ConvNext-XL [60] encoders and cooperates with other SOTA methods such as ELR [61], DivideMix [42], and UNICON [62]. However, our method outperforms EPL by achieving ∼6% higher accuracy on WebVision and ILSVRC12 datasets. This improvement over EPL demonstrates that developing better ways to incorporate pre-trained models to facilitate learning from noisy labels is a non-trivial task, highlighting the valuable contribution of our approach.

Table 3: Classification accuracies (%) on WebVision, ILSVRC2012 datasets.

| Dataset | DivideMix | ELR | UNICON | EPL | LongReMix | C2D | CC | NCR | LRA-diffusion |
|---|---|---|---|---|---|---|---|---|---|
| **WebVision** | 77.32 | 77.78 | 77.60 | 78.77 | 78.92 | 79.42 | 79.36 | 80.5 | **84.16** |
| **ILSVRC2012** | 75.20 | 70.29 | 75.29 | 76.51 | - | 78.57 | 76.08 | - | **82.56** |

Table 4: Classification accuracies (%) on the Food-101N dataset.

| Standard | CleanNet [57] | BARE [63] | DeepSelf [64] | PLC | LongReMix | LRA-diffusion |
|---|---|---|---|---|---|---|
| 81.67 | 83.95 | 84.12 | 85.10 | 85.28 | 87.39 | **93.42** |

For experiments on the Clothing1M dataset, we found that LRA-diffusion conditioned on the CLIP image encoder did not achieve the SOTA accuracy. A potential explanation is that the CLIP feature is too general for this domain specific task for categorizing fashion styles. However, our method is orthogonal to most traditional learning with noisy label approaches. As shown in the additional ablation study in Supplementary D.1, our method can collaborate with a trained classifier by conditioning on its feature encoder to achieve improved performance. We first use the CC [44] method to select clean samples and train a ResNet50 classifier, which achieved 75.32% accuracy (refer to as $CC^*$). Then, we condition on its feature before the classification head to train our LRA-diffusion model on the selected samples, which achieved 75.70% accuracy. As Table 5 shows, our method achieved a 0.38% improvement based on $CC^*$ and beat all SOTA methods.

Table 5: Classification accuracies (%) on Clothing1M

| Standard | BARE | PLC | LongReMix | DeepSelf | C2D | NCR | CleanNet |
|---|---|---|---|---|---|---|---|
| 68.94 | 72.28 | 74.02 | 74.38 | 74.45 | 74.58 | 74.60 | 74.69 |
| DivideMix | ELR | UNICON | EPL | $CC^*$ | CC | SANM [65] | LRA-diffusion |
| 74.76 | 74.81 | 74.98 | 75.21 | 75.32 | 75.40 | 75.63 | **75.70** |

## 5.4 Inference Efficiency Analysis

In order to test the efficiency of our model, we perform experiments assessing the runtime on CIFAR-10 dataset and compare our method with a standard classifier that uses ResNet50. It's worth noting that our SimCLR encoder is also built on the ResNet50. Thus, the standard method's runtime also reflects the linear probing runtime on SimCLR. Table 6 shows the results.

Table 6: Inference time (s) of standard classifier and LRA-diffusion models on CIFAR-10 images.

| Number of images | Standard (ResNet50) | LRA-diffusion | | |
|---|---|---|---|---|
| | | SimCLR (ResNet50) | CLIP (ViT-B/32) | CLIP (ViT-B/16) |
| 10000 | 3.96 | 9.52 | 9.13 | 17.12 |
| 50000 | 20.77 | 41.31 | 39.82 | 92.34 |

We can see, the computation bottleneck lies on the large pre-trained encoder but not the diffusion model itself. In general, our method takes twice as long as a standard classifier (ResNet50) when using SimCLR (ResNet50) and CLIP (ViT-B/32) pre-trained encoders. Larger CLIP encoders can increase the time further. However, it can be further accelerated if the features can be pre-computed in advance or be computed in parallel (as they are only required to be computed once and can be reused later).

## 6 Limitations

Our method, while being effective in many scenarios, does have certain limitations that we acknowledge. Its performance enhancement can be compromised if a pre-trained $f_p$ feature encoder isn't available or is inadequately trained. Additionally, the diffusion model introduces Gaussian noise in the forward process, leading to latent label vectors not being confined within the probability simplex, which could increase training time. Lastly, our method's performance becomes less effective when label noise levels surpass 50%. However, supervised learning can not be the optimal choice in such situations.

## 7 Conclusion

In this paper, by viewing the noisy labeling process as a conditional generative process, we leverage diffusion models to denoise the labels and accurately capture label uncertainty. A label-retrieval-augmented diffusion model was proposed to effectively learn from noisy label data by incorporating the principle of neighbor consistency. Additionally, by incorporating auxiliary conditional information from large pre-trained models such as CLIP, we are able to significantly boost the model performance. The proposed model is tested on several benchmark datasets, including CIFAR-10, CIFAR-100, Food-101N, and Clothing1M, achieving state-of-the-art results in most experiments. Future work could extend our model to multi-label settings, as it does not require a one-hot representation for labels. It is also promising to use semantic segmentation to guide the generation, potentially enhancing our model's interpretability and performance.

**Acknowledgement:** This work is partially supported by NSF AI Institute-2229873, NSF RI-2223292, an Amazon research award, and an Adobe gift fund. Any opinions, findings and conclusions or recommendations expressed in this material are those of the author(s) and do not necessarily reflect the views of the National Science Foundation, the Institute of Education Sciences, or the U.S. Department of Education.

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

# A  Denoising Diffusion Implicit Model with non-zero mean latent space

The forward process of a diffusion process with non-zero mean latent distribution $\mathbf{y}_T \sim \mathcal{N}(f(\mathbf{x}), \mathbf{I}))$ has a closed form representation:

$$q(\mathbf{y}_t|\mathbf{y}_0, f) = \mathcal{N}(\mathbf{y}_t; \sqrt{\bar{\alpha}_t}\mathbf{y}_0 + (1 - \sqrt{\bar{\alpha}_t})f(\mathbf{x}), (1 - \bar{\alpha}_t)\mathbf{I}, \tag{A.1}$$

which could be reprameterized as:

$$\mathbf{y}_t = \sqrt{\bar{\alpha}_t}\mathbf{y}_0 + (1 - \sqrt{\bar{\alpha}_t})f(\mathbf{x}) + \sqrt{1 - \bar{\alpha}_t}\epsilon \tag{A.2}$$

Then, similar to the DDIM, we define a Non-Markovian forward process with $\sigma_t \geq 0, t = 1:T$.

$$q_\sigma(\mathbf{y}_{1:T}|\mathbf{y}_0, f) := q_\sigma(\mathbf{y}_T|\mathbf{y}_0, f)\prod_{t=2}^{T} q_\sigma(\mathbf{y}_{t-1}|\mathbf{y}_t, \mathbf{y}_0, f) \tag{A.3}$$

Where $q_\sigma(\mathbf{y}_T|\mathbf{y}_0, f) = \mathcal{N}(\mathbf{y}_T; \sqrt{\bar{\alpha}_T}\mathbf{y}_0 + (1 - \sqrt{\bar{\alpha}_T})f(\mathbf{x}), (1 - \bar{\alpha}_T)\mathbf{I})$ and for all $t > 1$.

$$q_\sigma(\mathbf{y}_{t-1}|\mathbf{y}_t, \mathbf{y}_0, f) = \mathcal{N}\left(\mathbf{y}_{t-1}; \sqrt{\bar{\alpha}_{t-1}}\mathbf{y}_0 + (1 - \sqrt{\bar{\alpha}_{t-1}})f(\mathbf{x}) + \sqrt{1 - \bar{\alpha}_{t-1} - \sigma_t^2} \cdot \tilde{\epsilon}, \sigma_t^2\mathbf{I}\right), \tag{A.4}$$

$$\tilde{\epsilon} = \frac{1}{\sqrt{1 - \bar{\alpha}_t}} \cdot \left(\mathbf{y}_t - \sqrt{\bar{\alpha}_t}\mathbf{y}_0 - (1 - \sqrt{\bar{\alpha}_t})f(\mathbf{x})\right)$$

We can prove that the sampling process defined by Eq. (A.3) and Eq. (A.4) has the same *marginal distribution* as the closed-form sampling process in Eq. (A.1) by the following Lemma:

**Lemma A.1.** *For $q_\sigma(\mathbf{y}_{1:T}|\mathbf{y}_0, f)$ defined in Eq. (A.3) and $q_\sigma(\mathbf{y}_{t-1}|\mathbf{y}_t, \mathbf{y}_0, f)$ defined in Eq. (A.4), we have:*

$$q_\sigma(\mathbf{y}_t|\mathbf{y}_0, f) = \mathcal{N}(\mathbf{y}_t; \sqrt{\bar{\alpha}_t}\mathbf{y}_0 + (1 - \sqrt{\bar{\alpha}_t})f(\mathbf{x}), (1 - \bar{\alpha}_t)\mathbf{I}) \tag{A.5}$$

*Proof.* Assume for any $t \leq T$, if Eq. (A.5) is true, the following is also true:

$$q_\sigma(\mathbf{y}_{t-1}|\mathbf{y}_0, f) = \mathcal{N}(\mathbf{y}_{t-1}; \sqrt{\bar{\alpha}_{t-1}}\mathbf{y}_0 + (1 - \sqrt{\bar{\alpha}_{t-1}})f(\mathbf{x}), (1 - \bar{\alpha}_{t-1})\mathbf{I}), \tag{A.6}$$

then we can prove the statement with an induction argument for $t$ from $T$ to 1, since the base case $(t = T)$ already holds.
First, we have that:

$$q_\sigma(\mathbf{y}_{t-1}|\mathbf{y}_0) := \int_{\mathbf{y}_t} q_\sigma(\mathbf{y}_t|\mathbf{y}_0, f)q_\sigma(\mathbf{y}_{t-1}|\mathbf{y}_t, \mathbf{y}_0, f)d\mathbf{y}_t, \tag{A.7}$$

and

$$q_\sigma(\mathbf{y}_t|\mathbf{y}_0, f) = \mathcal{N}(\mathbf{y}_t; \sqrt{\bar{\alpha}_t}\mathbf{y}_0 + (1 - \sqrt{\bar{\alpha}_t})f(\mathbf{x}), (1 - \bar{\alpha}_t)\mathbf{I}) \tag{A.8}$$

$$q_\sigma(\mathbf{y}_{t-1}|\mathbf{y}_t, \mathbf{y}_0, f) = \mathcal{N}\left(\mathbf{y}_{t-1}; \sqrt{\bar{\alpha}_{t-1}}\mathbf{y}_0 + (1 - \sqrt{\bar{\alpha}_{t-1}})f(\mathbf{x}) + \sqrt{1 - \bar{\alpha}_{t-1} - \sigma_t^2} \cdot \tilde{\epsilon}, \sigma_t^2\mathbf{I}\right), \tag{A.9}$$

$$\tilde{\epsilon} = \frac{1}{\sqrt{1 - \bar{\alpha}_t}} \cdot \left(\mathbf{y}_t - \sqrt{\bar{\alpha}_t}\mathbf{y}_0 - (1 - \sqrt{\bar{\alpha}_t})f(\mathbf{x})\right).$$

According to [66] Eq. (2.115), we have that $q_\sigma(\mathbf{y}_{t-1}|\mathbf{y}_t, \mathbf{y}_0, f)$ is Gaussian, with mean $\boldsymbol{\mu}_{t-1}$ and co-variance $\boldsymbol{\Sigma}_{t-1}$:

$$\boldsymbol{\mu}_{t-1} = \sqrt{\bar{\alpha}_{t-1}}\mathbf{y}_0 + (1 - \sqrt{\bar{\alpha}_{t-1}})f(\mathbf{x}) \tag{A.10}$$
$$+ \sqrt{1 - \bar{\alpha}_{t-1} - \sigma_t^2}\left(\frac{\sqrt{\bar{\alpha}_t}\mathbf{y}_0 + (1 - \sqrt{\bar{\alpha}_t})f(\mathbf{x}) - \sqrt{\bar{\alpha}_t}\mathbf{y}_0 - (1 - \sqrt{\bar{\alpha}_t})f(\mathbf{x})}{\sqrt{1 - \bar{\alpha}_t}}\right)$$
$$= \sqrt{\bar{\alpha}_{t-1}}\mathbf{y}_0 + (1 - \sqrt{\bar{\alpha}_{t-1}})f(\mathbf{x}). \tag{A.11}$$

$$\boldsymbol{\Sigma}_{t-1} = \sigma_t^2\mathbf{I} + \frac{1 - \bar{\alpha}_{t-1} - \sigma_t^2}{1 - \bar{\alpha}_t}(1 - \bar{\alpha}_t)\mathbf{I} = (1 - \bar{\alpha}_t)\mathbf{I}. \tag{A.12}$$

Therefore, Eq. (A.6) holds. Following the induction, the lemma is proved. $\qquad\square$

In our implementation, we follow DDIM [32] setting $\sigma_t = 0$. The resulting model becomes an implicit probabilistic model [67], where the generation process become deterministic given $\mathbf{y}_T$.

# B    t-SNE visualization of the DDIM generation process

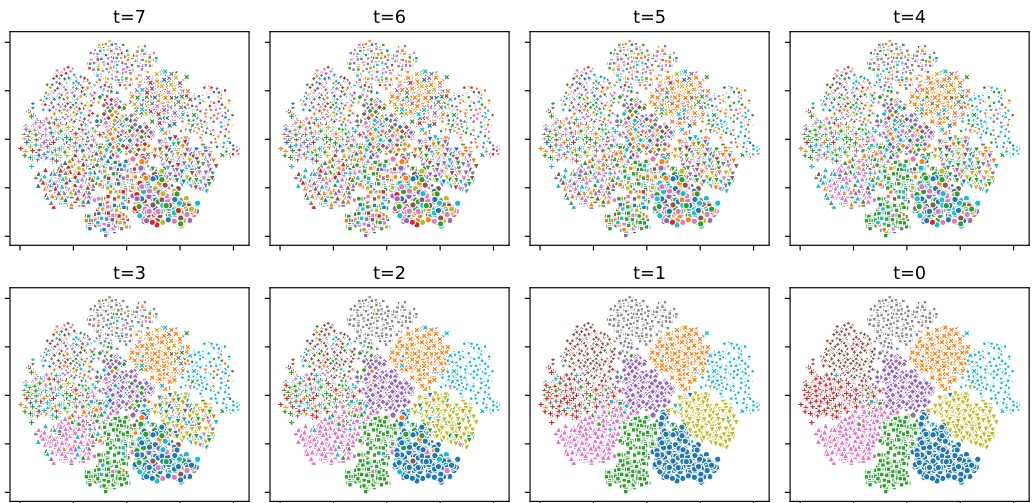

Figure B.1: The t-SNE visualization of the CLIP feature space during the reverse generation process of a conditional diffusion model using an 7-step DDIM on the CIFAR-10 dataset. The process begins at time $t = 7$, with the sampling of the latent representation of the label from the latent distribution $\mathcal{N}(f_q(\mathbf{x}), \mathbf{I})$. Through a series of multi-step reverse operations, the latent distribution is transformed into the conditional distribution of labels. The data points are color-coded according to the entry with the highest value in intermediate/final label vectors, and the ground truth class labels are represented by distinct markers.

# C    Experimental setup and details

## C.1    Real-world dataset details

**WebVision** comprising 2.4 million images that were crawled using Google and Flickr search engines, with the ILSVRC12 taxonomy. Following prior studies, we trained our model on the initial 50 classes from the Google image subset of Webvision and tested it on the validation sets of both Webvision and ILSVRC12.

**Food-101N** consists of 310k food images collected from the internet with the Food-101 [68] taxonomy, and has an estimated label noise level of 20%, making it an ideal dataset to evaluate the robustness of our method under real-world noisy labels. We assessed the classification accuracy on the curated label set of Food-101, which contains around 25k images.

**Clothing1M** contains 1 million images of clothes obtained from shopping websites. Based on the keywords in the surrounding text, the images are automatically classified into 14 classes with ∼40% estimated noise level. The dataset includes a clean training set, validation set, and test set with manually refined labels, consisting of approximately 47.6k, 14.3k, and 10k pictures, respectively. We discarded the clean training set and only used the noisy label data for training.

## C.2    Implementation details

To present the hyperparameter settings of our neural network, we first give a description of our neural network design. As shown in Figure C.1, the network consists of a frozen $f_p$ encoder, a ResNet encoder, and a series of feed forward layers. Features encoded by the two encoders are combined with time embedding via hadamard product and passed through a series of feed-forward networks, batch normalization, and softplus activation to predict the noise term $\epsilon_\theta$.

In our experiments, we use ResNet34 for CIFAR10 and CIFAR100, and use ResNet50 for real-world datasets as the trainable encoder (blue ResNet block in Figure C.1). The dimensions of all feed-forward layers are set to 512 for CIFAR datasets and 1024 for real-world datasets, respectively. We train LRA-diffusion models for 200

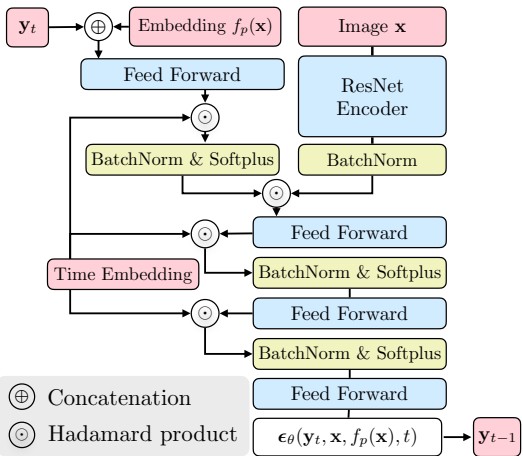

Figure C.1: The network architecture for conditional diffusion models. The input to the network consists of four elements: $\mathbf{y}_t$, $f_p(\mathbf{x})$, $\mathbf{x}$, and the time embedding for t, represented by pink blocks. The blue blocks in the figure represent the trainable network components.

epochs with Adam optimizer. The batch size is 256. We used a learning rate schedule that included a warmup phase followed by a half-cycle cosine decay. The initial learning rate is set to 0.001. Following [15], we applied data augmentation in the training, including resizing, random horizontal flip, and random cropping. To retrieve the nearest neighbors, we set k=10 based on our tests using a range of k values from 1 to 100 on the validation sets. The KNN accuracy remained relatively stable for k between 10 and 50, and then starts to decline due to reduced label consistency among neighbors. Based on these results, we infer that our LRA diffusion model is less sensitive to variations in $k$ within this range. All experiments are conducted using four NVIDIA Titan V GPUs.

## D    Additional ablation study

### D.1    Classifier feature conditioning for accuracy enhancement

To demonstrate how our method can enhance a trained classifier's performance by using its features as conditional information, we conduct an ablation study examining the impact of conditional diffusion and KNN on the trained classifier. Specifically, we train classifiers, denoted as $\eta(\mathbf{x})$, using the standard method at various noise levels. We then remove the classification head and utilize the remaining model $f_\eta$ as the $f_p$ encoders in our conditional diffusion models.

The experimental results shown in Figure D.1 indicate that these techniques can improve test accuracy. We observe that when the noise level is below $55\%$, the conditional diffusion model (green) achieves a $\sim 1\%$ improvement over the standard method. Moreover, when the LRA method is applied concurrently (purple), test accuracy can be further enhanced. This improvement occurs because learning from neighbors' labels reduces the noise level during training, as evidenced by the comparison between KNN results (blue) and clean label percentage (gray).

However, when the noise level exceeds $55\%$, the use of diffusion and LRA-diffusion methods does not seem advantageous. This limitation arises because the distribution of labels in the neighborhood becomes too corrupted for KNN to effectively improve the proportion of clean labels during training, as illustrated by the intersection of the blue and gray curves in the figure. We argue this does not diminish the practical value of our method because a dataset with more than 50% label noise is not meaningful in practice.

### D.2    Effects of pseudo-label construction strategies

We also conduct comparative experiments using another method that utilizes neighbor labels: replacing the one-hot label vector with the mean vector of the neighbor's labels as the prediction target, which we call Mean-diffusion. We found that it can achieve higher accuracy when the noise level is higher than $55\%$. This may be due to the increase in the diversity of neighbor labels. The sampling-based LRA-diffusion will need

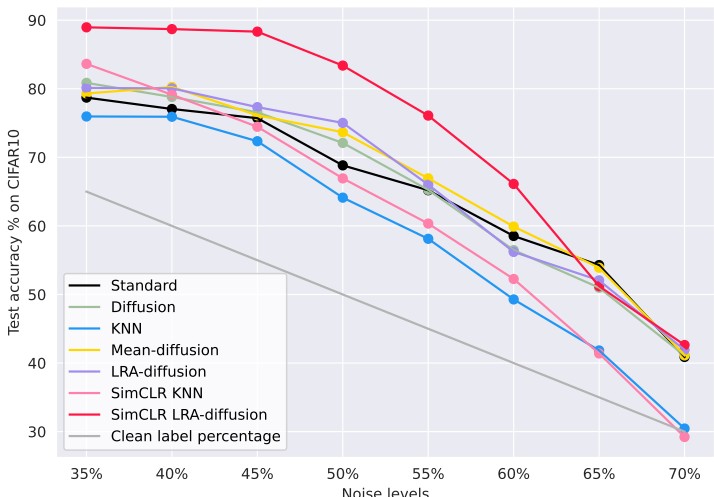

Figure D.1: Test accuracy of seven methods on the CIFAR-10 dataset with different levels of PMD label noise. Other than the already introduced method names, here *Diffusion* is a conditional diffusion model using the feature $f_\eta$. The clean label percentage is represented by the gray line.

to learn a more complex multi-modal distribution, but Mean-diffusion only needs to learn a point estimate. However, when the noise level is lower than $55\%$, we found that LRA-diffusion is slightly more accurate than Mean-diffusion. A possible explanation is that the distribution of $\mathbf{y}_0$ in LRA-diffusion contains only n one-hot labels. In contrast, $\mathbf{y}_0$ in Mean-diffusion is more diverse ($n^k/k!$ possible mean vectors for $n$ classes and $k$ neighbors). In conclusion, LRA-diffusion has higher performance with less noisy labels. On the other hand, Mean-diffusion has faster and more stable convergence and is more robust for high noise level. However, they tend to perform similarly when the noise level is too high or too low since neighbors' labels will become the same or too corrupted.

## D.3 Robustness of SimCLR feature conditioning

Finally, we use the SimCLR model as the encoder $f_p$ in our conditional diffusion model (listed as SimCLR LRA-diffusion in Figure D.1), to showcase the effectiveness of our proposed LRA-diffusion method in utilizing prior knowledge from pre-trained image representations to enhance the test accuracy and robustness. The experimental results (red) show that its test accuracy significantly surpasses other settings until the noise level reaches $65\%$. Beyond this point, the labels in the neighborhood become too corrupted to provide additional supervision information.

## D.4 Effects of CLIP feature conditioning strategies

Table D.1: Classification accuracy (%) of linear probing, KNN, and diffusion model using pre-trained CLIP feature on real-world label noise datasets.

| Method | Webvision | ILSVRC2012 | Food-101N | Clothing1M |
|---|---|---|---|---|
| KNN | 81.88% | 82.12% | 91.73% | 65.70% |
| linear prob + Mean label | 68.56% | 68.76% | 90.68% | 65.75% |
| linear prob + sample label | 54.84% | 56.80% | 89.56% | 55.52% |
| diffusion (Mean label) | 83.96% | 82.24% | 93.13% | **71.79%** |
| diffusion (sample label) | **84.16%** | **82.56%** | **93.42%** | 71.65% |

We carried out an ablation study employing various strategies to integrate the CLIP model for classification on real-world datasets. The results demonstrate that the superior performance of our diffusion-based method does not simply reflect the strength of the CLIP features. We show results of linear probing and KNN using CLIP features in Table D.1 to further demonstrate the effectiveness of our algorithm design. The results suggested that

utilizing linear probing with LRA and mean labels degrades the pre-trained feature space, leading to diminished performance in comparison to the unsupervised KNN approach. On the other hand, the diffusion process can effectively incorporate the CLIP feature space to achieve higher performance.

