# OpenReview forum: "Label-Retrieval-Augmented Diffusion Models for Learning from Noisy Labels"
_NeurIPS.cc/2023/Conference — NeurIPS 2023 poster_

### Official Review · Reviewer_poWF · 2023-07-06

**Soundness:** 3 good
**Presentation:** 3 good
**Contribution:** 3 good
**Rating:** 6
**Confidence:** 4

**Summary:**

This paper proposes a Label-Retrieval-Augmented (LRA) diffusion model for learning from noisy labels. The model leverages the neighbor consistency principle and incorporates pre-trained models to improve performance. The paper introduces the label-retrieval-augmented component, an accelerated label diffusion process, and a new conditional mechanism for incorporating pre-trained models. The proposed model achieves good performance both on synthetic and real-world benchmark datasets and the experimental results show that it can boost accuracy by 10-20 absolute points in many cases.

**Strengths:**

1.The idea of using the diffusion model to address the noisy label learning problem is interesting.

2.Extensive experimental results demonstrate the proposed method can achieve the state-of-the-art method compared with multiple peer methods.

**Weaknesses:**

The experiment part needs to be improved: add state-of-the-art method, discuss the influence of the label noise type, and the influence of different pre-trained models.

**Questions:**

1. The paper mentions that the proposed model is flexible and general, but it does not provide a detailed discussion of the types of label noise that the model can effectively handle. For example, feature-dependent label noise. Since the comparison method [15] is a baseline that focuses on feature-dependent label noise.

2. It seems the pre-trained model is important for the proposed method. It would be better to discuss the effect of different backbone networks as well as pre-trained model on model performance.

3. The description of the comparison algorithms is lacking in the paper.

4. In Table 1, the comparison methods are outdated. It would be better to add some state-of-art methods published in 2022 or 2023.

**Limitations:**

Please refer to the weakness and limitations.

---

> ### Author Rebuttal · Authors · 2023-08-10
>
> Q1: The paper mentions that the proposed model is flexible and general, but it does not provide a detailed discussion of the types of label noise that the model can effectively handle. For example, feature-dependent label noise. Since the comparison method [15] is a baseline that focuses on feature-dependent label noise.\
> A: Following PLC [15], we conduct experiments using their settings on polynomial margin diminishing (PMD) noise, a novel class of synthetic feature-dependent noise (synonyms of instance-dependent noise). This noise is more realistic to real-world label noise and more difficult to handle than i.i.d noise (instance-independent). However, our method can handle instance-dependent noise, since our model approximates the distribution of neighboring labels of each instance, allowing it to make instance-dependent predictions. Results shown in Table 1 demonstrated that our method with the SimCLR feature achieved state-of-the-art results on the PMD noise. We will provide a clearer explanation in the revised manuscript.
>
> Q2: It seems the pre-trained model is important for the proposed method. It would be better to discuss the effect of different backbone networks as well as pre-trained models on model performance.\
> A: We agree that the network design of diffusion models is an interesting topic. However, the backbone we adopted from the CARD model for 1D diffusion (supplementary figure C.1) is already very efficient and effective, and there are no well-known alternative designs. Thus, the impact of different backbone networks is hard to evaluate. In addition, we believe the quality of the feature has a positive relationship with the classification accuracy.
>
> Q3: The description of the comparison algorithms is lacking in the paper.\
> A: Thank you for noting the lack of detail in our comparison algorithms. We will add the required information to the supplementary materials in the revised manuscript.
>
> Q4: In Table 1, the comparison methods are outdated. It would be better to add some state-of-art methods published in 2022 or 2023.\
> A: Table 1 reports the results on synthetic label noise, specifically focusing on the polynomial margin diminishing (PMD)  noise, which is a new class of synthetic instance-dependent label noise. Recent state-of-the-art (SOTA) methods have not performed evaluations on PMD noise in their original papers. To address the problem, we additionally perform experiments using the source code of two SOTA methods, C2D (2022) and CC (2022) and show the results in Table 1 of the attached PDF. C2D also utilizes a pre-trained SimCLR encoder for initialization, but label noise may still affect the feature space during training. In contrast, our method freezes the feature encoder, shielding the pre-trained features from noise. Results on CIFAR-10 demonstrate that when the pre-trained feature is of high quality, our method achieves superior accuracy. On CIFAR-100, where the SimCLR feature has lower KNN accuracy, C2D is more effective, as it can refine the feature space through training. Freezing the feature encoder has another advantage: it enables us to efficiently incorporate more powerful pre-trained encoders, such as CLIP. This approach spares us from the prohibitive computational burden of fine-tuning the feature encoder, allowing for more effective integration.\
> On the other hand, we have included more recent SOTA baselines in experiments on real-world datasets. This inclusion is of greater importance, as it reflects performance in real-world applications, providing a more practical and relevant assessment of our method.

---

> > ### Comment · Reviewer_poWF · 2023-08-16
> >
> > Thanks for the authors’ response.
> >
> > **To A4**, the author only uses the SOTA method to validate the effectiveness of the proposed method on ILSVRC2012 and Food-101N but does not give a validation on the commonly used datasets CIFAR-10 and CFIAR-100.
> >
> > **To A1** The author does not answer my question completely. Can the author provide the comparison result on the label noise produced by the baseline PLC?

---

> > > ### Author Response · Authors · 2023-08-17
> > >
> > > **To A1**, The author does not answer my question completely. Can the author provide the comparison result on the label noise produced by the baseline PLC?
> > >
> > > **R1**: We apologize for any confusion. Yes, our method can handle the noise produced by PLC (PMD noise). The results in Table 1 of our manuscript are exactly on the label noise produced by PLC. We downloaded the noisy labels from PLC's GitHub repository, and all the results were borrowed directly from PLC's original paper.
> > >
> > > **To A4**, the author only uses the SOTA method to validate the effectiveness of the proposed method on ILSVRC2012 and Food-101N but does not give a validation on the commonly used datasets CIFAR-10 and CFIAR-100.
> > >
> > > **R4**: We have included results on CIFAR-10 and CIFAR-100 for two SOTA methods (C2D and CC) in Table 1 of the attached PDF in the global response. These methods achieved the first and second-highest accuracy on ILSVRC2012. Since other methods (NCR and SANM) did not provide complete source code for training, we couldn't produce their results during the rebuttal. However, we will include more SOTA baselines in the final version.

---

> > > > ### Comment · Reviewer_poWF · 2023-08-17
> > > >
> > > > Thanks for your response.
> > > >
> > > > In fact, the global response from my end does not include any additional attached PDF.

---

> > > > > ### Author Response · Authors · 2023-08-17
> > > > >
> > > > > There’s a very small link for the PDF at the bottom left corner of the global response. It is hard to spot, but it should be available to reviewers.
> > > > >
> > > > > Since it is not an external link, I think I am allowed to paste it here for your convenience:
> > > > > https://openreview.net/attachment?id=7cav4HZB23&name=pdf

---

> > > > > > ### Comment · Reviewer_poWF · 2023-08-17
> > > > > >
> > > > > > Thanks for your response.
> > > > > >
> > > > > > I hold a concern regarding the experimental comparison, which primarily focuses on self-assessment using different pre-trained models, wherein the observed performance improvement appears to be largely attributed to the utilization of a stronger pre-trained model.
> > > > > >
> > > > > > Furthermore, the comparison with the SOTA method can be biased due to the incorporation of a highly robust pre-trained model into the proposed approach.
> > > > > >
> > > > > > In the ablation study, could you elucidate the reasons behind the superior performance of Linear probing on SimCLR as compared to Linear probing on SimCLR+LRA?

---

> > > > > > > ### Author Response · Authors · 2023-08-18
> > > > > > >
> > > > > > > **Q1**: I hold a concern regarding the experimental comparison, which primarily focuses on self-assessment using different pre-trained models, wherein the observed performance improvement appears to be largely attributed to the utilization of a stronger pre-trained model.
> > > > > > > Furthermore, the comparison with the SOTA method can be biased due to the incorporation of a highly robust pre-trained model into the proposed approach.
> > > > > > >
> > > > > > > **R1**: Thank you for your insightful observation that performance improvement is mainly attributed to using a pre-trained model. In fact, integrating pre-trained models in learning from noisy labels is actively being explored due to the availability of the CLIP encoder and the significant boost in the performance it can bring, such as EPL (Jongwoo Ko, 2023). However, effectively incorporating such features to assist the learning from noisy labels (LNL) is a non-trivial task. Our method achieved ~6% higher accuracy on WebVision and ILSVRC12 than EPL, which incorporates the most powerful CLIP and ConvNext-XL encoders as well as multiple SOTA LNL methods. This additional improvement compared to EPL is a valuable contribution of our diffusion-based method. We will add this comparison in our final version.
> > > > > > > * Ko, Jongwoo, Sumyeong Ahn, and Se-Young Yun. "EFFICIENT UTILIZATION OF PRE-TRAINED MODEL FOR LEARNING WITH NOISY LABELS." ICLR 2023 Workshop on Pitfalls of limited data and computation for Trustworthy ML. 2023.
> > > > > > >
> > > > > > > **Q2**: In the ablation study, could you elucidate the reasons behind the superior performance of Linear probing on SimCLR as compared to Linear probing on SimCLR+LRA?
> > > > > > >
> > > > > > > **R2**: Our results show that this performance gap is specific to using sampled labels and does not occur with mean labels. This difference may be due to the linear layer's inability to yield stochastic outputs from a multimodal distribution. During training, conflicting gradient directions may arise if the model tries to predict different labels across gradient steps, ultimately impeding learning. However, the diffusion model can effectively learn to generate different outputs. The use of neighboring labels only affects the probability of generating each label.
> > > > > > > We also observed that this phenomenon was only observed with SimCLR and not with CLIP. One possible explanation might be the robustness of CLIP features, which appear to mitigate the effects of conflicting gradient directions induced by different labels, thus preventing any significant hindrance to learning.

---

> > > > > > > > ### Comment · Reviewer_poWF · 2023-08-18
> > > > > > > >
> > > > > > > > Thanks for your further explanation. I will raise my score.

---

### Official Review · Reviewer_ZUT9 · 2023-07-07

**Soundness:** 2 fair
**Presentation:** 3 good
**Contribution:** 2 fair
**Rating:** 5
**Confidence:** 3

**Summary:**

The paper focuses on learning from noisy labels using diffusion models. They reformulate the noisy label problem from a generative perspective. Specifically, the clean label is our target variable ($y_0$) and the noisy label is the diffused variable ($y_T$), which they call Label-Retrieval-Augmented (LRA) diffusion model. Actually, we don't know the clean label, they extract the pseudo clean labels using the neighborhood consistency.

**Strengths:**

* The paper is well-written.
* The proposed method is a well-extended approach that builds upon the utilization of diffusion models in classification tasks and applies it effectively to tackle the noisy label problem.

**Weaknesses:**

* Motivation of LRA diffusion model
  * It is important to explain why the noisy label process should be formulated as a diffusion process and to discuss in detail the advantages it offers.
  * Since I think it is a novel approach, it is crucial to provide detailed explanations of the motivation behind it.
  * When using the neighbor consistency principle, it is important to address whether the same problems mentioned by the authors in lines 44-51 would arise. If not, it is necessary to explain how the proposed model overcomes these challenges.

* Comparison with CARD
  * It appears that the original CARD paper introduced two pre-trained encoders, while the current paper seems to have used only one. This distinction needs to be clarified, and the roles of the two encoders should be explained separately in the description.
  * Equation 4 seems to be adapted from the formulation presented in CARD, and proper citation should be provided.

* Experiments
  * The utilization of CLIP in the performance evaluation raises concerns because it involves the incorporation of additional data. It is important to conduct a fair comparison by addressing this aspect.
  * An explanation is needed to clarify why the diffusion model is not a computational bottleneck. It is hypothesized that this may be due to the smaller dimensions, but this should be verified.

* Minor comments
  * The appearance of $\tilde{y}$ in line 136 without prior definition should be addressed.
  * "Require: Input:" in Algorithm 1 is typo.
  * The inconsistent use of $\mu_{\theta}$ and $\epsilon_{\theta}$ should be corrected.
  * The subscript in line 153 needs to be written correctly.

**Questions:**

Please see Weaknesses part.

**Limitations:**

They discussed in Limitations section.

---

> ### Author Rebuttal · Authors · 2023-08-10
>
> Q1.1: It is important to explain why the noisy label process should be formulated as a diffusion process and to discuss in detail the advantages it offers. Since I think it is a novel approach, it is crucial to provide detailed explanations of the motivation behind it.\
> A: Thank you for highlighting the need to explain our approach. The intrinsic ambiguity of data introduces uncertainty to the labeling process, resulting in controversial labels. Modeling this process with a stochastic conditional generative model is intuitive, and using the most probable generation is expected to improve the annotation (classification) accuracy. We chose the diffusion model because it has demonstrated the ability to model prediction uncertainty in the CARD model for classification. In addition, the mode coverage ability of the diffusion model (generating diverse samples given the same conditional information) is intuitively suitable for training with neighboring labels. We'll provide more detailed explanations in the revised manuscript.
>
> Q1.2: When using the neighbor consistency principle, it is important to address whether the same problems mentioned by the authors in lines 44-51 would arise. If not, it is necessary to explain how the proposed model overcomes these challenges.\
> A: In our method, the feature encoder is fixed and does not change during the training process. As a result, the feature space is determined prior to the introduction of noisy labels and is not subject to distortion by label noise.
>
> Q2.1: It appears that the original CARD paper introduced two pre-trained encoders, while the current paper seems to have used only one. This distinction needs to be clarified, and the roles of the two encoders should be explained separately in the description.\
> A: The original CARD paper only introduced one pre-trained encoder, which is used as the mean estimator of the diffusion start at $t=T$. This encoder's features were also used as input to CARD's neural network. However, due to its role as the mean estimator, the encoder was restricted to a low dimension (the number of classes), thereby limiting its representation capacity. Intuitively, one might expect that feeding higher-dimension, more powerful features to the network could lead to performance improvements.\
> Therefore, in our method, we introduced a separate feature encoder specifically to feed conditional information to the network. We denote the mean estimator as $f_q$ (the same dimension as the labels), and the feature encoder (higher dimension) as $f_p$. As a result, the CARD model can be seen as a special case of our model, where $f_p = f_q$. In Table 2, we show CARD+LRA results in poor performance in our ablation study. This design is explained in section 3.4, lines 170-175 of our manuscript. We apologize if this was not clear in our initial presentation, and we will certainly make efforts to clarify this in the revised manuscript.
>
> Q2.2: Equation 4 seems to be adapted from the formulation presented in CARD, and proper citation should be provided.\
> A: Thanks for pointing out this. We have updated our manuscript with the appropriate reference to the CARD work.
>
> Q3.1: The utilization of CLIP in the performance evaluation raises concerns because it involves the incorporation of additional data. It is important to conduct a fair comparison by addressing this aspect.\
> A: Please refer to the global response. We have added additional results in Table 3 of the attached PDF file, where simply applying the CLIP features leads to poor performance. We hope this response addresses your concern.
>
> Q3.2: An explanation is needed to clarify why the diffusion model is not a computational bottleneck. It is hypothesized that this may be due to the smaller dimensions, but this should be verified.\
> A: As depicted in the supplementary figure C.1, all the time embeddings are inserted into the feed-forward layers of the neural network. Thus, the computation of the ResNet encoder is required only once and is the same as that for a standard ResNet classifier. The additional computation cost involves the computation of the $f_p$ encoder and a label-dimension (e.g., 10 in CIFAR10) diffusion process, which is quite efficient. A qualitative comparison in Table 6 shows that the time cost of our method using a ResNet50 as $f_p$ encoder is comparable to a standard ResNet50 classifier, while the performance is much better.
>
> Minor Comments\
> A: Thank you for identifying the writing errors in our manuscript. We appreciate your attention to detail. We will correct these mistakes and do proofreading to ensure that no further errors remain.

---

> > ### Comment · Reviewer_ZUT9 · 2023-08-16
> >
> > Thank you for the author's response. I have addressed most of the concerns, but I still have a question.
> >
> > **Q1.2**
> >
> > My question is that this fixed feature encoder might introduce the problem as the author said (*The performance highly depends on the quality of the encoder that maps the data to the feature space, ... the training can also lead to overfitting or underfitting.*).

---

> > > ### Author Response · Authors · 2023-08-17
> > >
> > > **R**: Thank you for raising this critical point. As you pointed out, a low-quality fixed feature encoder will not benefit the learning. However, when the pre-trained feature is of high quality, our method is more effective than fine-tuning the feature space using the SOTA method (C2D) because it shields the feature space from distortion by label noise. Moreover, freezing the feature encoder enables the integration of more powerful pre-trained encoders, such as CLIP, because it frees us from the prohibitive computational burden of fine-tuning.
> > > For a detailed comparison, please refer to the new baseline results, 'C2D + SimCLR', in Table 1 of the attached PDF. C2D employs a pre-trained SimCLR encoder for initialization; thus, the weight is fine-tuned during training. Results on CIFAR-10 demonstrate that when the pre-trained feature is of high quality, our method achieves superior accuracy. On CIFAR-100, where the SimCLR feature has lower quality (lower KNN accuracy), C2D is more effective, as it can refine the feature space through training.

---

> > > > ### Comment · Reviewer_ZUT9 · 2023-08-18
> > > >
> > > > Thanks for your extra explanation. After reading the rebuttal and other reviews, I would like to raise my rating from 4 to 5.

---

### Official Review · Reviewer_3Gk3 · 2023-07-07

**Soundness:** 3 good
**Presentation:** 4 excellent
**Contribution:** 3 good
**Rating:** 5
**Confidence:** 3

**Summary:**

The paper proposes the application of denoising diffusion probabilistic models for modeling the true class probability distribution when training with noisy labels. In particular, the paper extends Classification and Regression Diffusion Models for this problem and uses pretrained self-supervised representation models for Label Retrieval Augmentation. At test time, they approximate the true label by starting with the reverse diffusion at the mean of the final gaussian distribution. The empirical results on CIFAR10, CIFAR100 and other real-world noisy datasets show improvements over considered baselines.

**Strengths:**

The main strength of this work is that it demonstrates successful application of diffusion models for learning the probability distribution of labels in noisy label settings. The paper considers a variety of datasets and includes the relevant ablation studies and inference times.

**Weaknesses:**

I believe the key weaknesses are:
1) The requirement of a pre-trained self-supervised image embedding network.
2) The diffusion model does not sample discrete labels and does not even model the categorical distribution.
3) I think the evaluation should also consider the following baseline: train a classifier network exactly as in Algorithm 1 except in step 5 where the network is trained with an MSE loss to predict one-hot vector obtained in step 4. More specifically, it would be good to check the performance when the classifier network can take in the image and the simclr/clip features and have a linear prediction head. This baseline would be more closer to the LRA-diffusion model than the other baselines considered in Table 2 in terms of the model input and output.

**Questions:**

1. In line 179, you write that you simply set $f_q({\bf x}) = {\bf 0}$ whereas you write in line 151 that the DDIM generation begins with a non-zero centered gaussian. So, the generalized DDIM is essentially used for the CARD+LRA Diffusion experiments in Table 2? And, you have a zero-centered gaussian distribution for the LRA-diffusion that you actually implement?
2. Do you have any intuition about the performance of the baseline proposed in Weakness (3)?
3. Do the baselines considered in Tables (3), (4) and (5) have access to extra training data?
4. Will you release all the trained model checkpoints used for reporting the results in tables?


**Limitations:**

Yes, the limitations are adequately addressed.

---

> ### Author Rebuttal · Authors · 2023-08-10
>
> W1: The requirement of a pre-trained self-supervised image embedding network.\
> A: We believe the requirement of a self-supervised embedding model like SimCLR is not necessarily a limitation, as it can be obtained for free by training on the same training dataset. Furthermore, the utilization of SimCLR is a common practice in the field. It has been leveraged in previous state-of-the-art methods such as C2D and the methods proposed by (Smart
> 2023 and Cordeiro 2021).
> * Smart, Brandon, and Gustavo Carneiro. "Bootstrapping the Relationship Between Images and Their Clean and Noisy Labels." Proceedings of the IEEE/CVF Winter Conference on Applications of Computer Vision. 2023.
> * Cordeiro, Filipe R., et al. "Propmix: Hard sample filtering and proportional mixup for learning with noisy labels." arXiv preprint arXiv:2110.11809 (2021).
>
> W2: The diffusion model does not sample discrete labels and does not even model the categorical distribution. \
> A: We would like to argue that this is not a fundamental issue, as the generated label vector can be interpreted as the logit of a prediction head of a traditional classifier. To further address your issue, we also performed experiments using diffusion on the probability simplex following the Star-shaped DDPM (Okhotin 2023). As stated in their paper, the utilization of Dirichlet distribution (the conjugate prior of the categorical distribution) in a diffusion mode requires a different mechanism design than DDPM. We have added the results as Dirichlet diffusion in Table 2. Although it has a one-hot label interpretation, the performance is indeed slightly worse than using standard DDPM. We hope this response adequately addresses your concern.
>
> * Okhotin, Andrey, et al. "Star-Shaped Denoising Diffusion Probabilistic Models." arXiv preprint arXiv:2302.05259 (2023).
>
> W3 and Q2: I think the evaluation should also consider the following baseline: train a classifier network exactly as in Algorithm 1 except in step 5 where the network is trained with an MSE loss to predict one-hot vector obtained in step 4. More specifically, it would be good to check the performance when the classifier network can take in the image and the simclr/clip features and have a linear prediction head. This baseline would be more closer to the LRA-diffusion model than the other baselines considered in Table 2 in terms of the model input and output.\
> A: We have included the results of this baseline, denoted as "ResNet + linear," in Table 2 of the attached PDF file, and plan to incorporate it into the final version of the manuscript. We used a similar network architecture and the same pre-trained features and training labels as those used for our diffusion model.  In certain settings, its accuracy is found to be lower than that of linear probing without using images as input. This difference may be attributed to the additional learning capacity introduced by the ResNet component being used to overfit the label noise. Moreover, the performance gap between our method and this baseline demonstrates the significance of the diffusion process.
>
> Q1: In line 179, you write that you simply set whereas you write in line 151 that the DDIM generation begins with a non-zero centered gaussian. So, the generalized DDIM is essentially used for the CARD+LRA Diffusion experiments in Table 2? And, you have a zero-centered gaussian distribution for the LRA-diffusion that you actually implement? \
> A: You are correct. The non-zero centered Gaussian for the start of the generation process is an essential component introduced in CARD. We also include this design in our model to present a more theoretically general framework of our method. However, in practice, we must adopt the setting that is most effective. Through our empirical analysis, we found that utilizing a noisy classifier as the $f_q$ encoder does not consistently lead to improved results. For the sake of simplicity and to reduce model complexity, we thus chose to use only a zero mean.
>
> Q3: Do the baselines considered in Tables (3), (4) and (5) have access to extra training data? \
> A: The baseline methods do not have access to extra training data. In the global response, we discussed that the performance of our method does not simply reflect the strength of the CLIP features by comparing our method to the KNN and linear probing results using CLIP features on the real-world datasets presented in Tables 3, 4, and 5.
>
> Q4: Will you release all the trained model checkpoints used for reporting the results in tables?\
> A: Yes, we will release all checkpoints and the code to recreate the results of our experiments.

---

> > ### Comment · Reviewer_3Gk3 · 2023-08-13
> > **Response to Rebuttal.**
> >
> > Thank you for the extra experiments and answers to my questions.

---

### Official Review · Reviewer_21mt · 2023-07-07

**Soundness:** 2 fair
**Presentation:** 2 fair
**Contribution:** 2 fair
**Rating:** 3
**Confidence:** 4

**Summary:**

The paper proposes using a diffusion model to perform classification on a noisy-label dataset. The authors utilize diffusion to learn how to conditionally generate labels given an image. By combining recent work on classification diffusion models with label retrieval augmentations they demonstrate state-of-the-art results on real-world noisy datasets.

**Strengths:**

- The method provides a simple, yet seemingly effective way of performing classification on noisy data using generative models. The idea of using diffusion models for purpose this is novel and has the potential to significantly contribute to the "Learning With Label Noise" literature.
- Some of the real-world dataset results showcased outperform the previous state-of-the-art by a significant margin.

**Weaknesses:**

- The novelty of the paper is limited. The authors combine the existing classification diffusion model with the retrieval augmentation technique to capture the noisy label distribution. The methodologies for both are inherited almost unchanged from their respective papers and there is no significant extension to either. Extending DDIM to a non-zero mean latent distribution is a minor contribution.

- In lines 127 and 166, the authors claim that initializing with the predicted label mean and running the DDIM sampler approximates the maximum likelihood estimation of the labels. This is not justified anywhere, and there is no evidence to support the claim.

- The experiments on the WebVision and ILSVRC2012 datasets should be examined further. The method presented in this paper relies heavily on having a high-quality pre-trained encoder network that can provide meaningful representations during training. For these two experiments, the authors utilized CLIP, whereas the next best-performing approaches, C2D and NCR, were only trained using samples from the smaller datasets. It is possible that the majority of the performance increases shown in this paper can be attributed to the discriminative ability of the CLIP encoder, since additionally on the Clothing1M dataset, where CLIP is not expected to perform well, the authors did not manage to greatly outperform the previous state-of-the-art results.


**Questions:**

- For a single sample, is the frequency of the neighbors seen in training the same regardless of their distances in the latent space? If so, how is a neighborhood defined?

**Limitations:**

The limitations are addressed. Broader societal impacts are not applicable.

---

> ### Author Rebuttal · Authors · 2023-08-10
>
> W1: The novelty of the paper is limited. The authors combine the existing classification diffusion model with the retrieval augmentation technique to capture the noisy label distribution. The methodologies for both are inherited almost unchanged from their respective papers and there is no significant extension to either. Extending DDIM to a non-zero mean latent distribution is a minor contribution.\
> A: Thanks for the comment. We wish to claim that there are at least two types of research papers: one focuses on theoretical innovation; the other focuses on improving practical performance of some particular problems. Our paper largely belongs to the second type. Our contribution lies in integrating and adapting diffusion models and the SOTA image encoder to a specific problem domain rather than proposing a new theoretical analysis. The proposed design allows us to achieve significant performance improvements, as demonstrated in our experimental results. On the other hand, extending DDPM to a non-zero mean latent distribution is essential to the previous CARD model. In our paper, we extended it to DDIM to provide a more general framework that includes CARD as a special case and potentially facilitates future work that uses a more refined $f_q$ encoder.
>
> W2: In lines 127 and 166, the authors claim that initializing with the predicted label mean and running the DDIM sampler approximates the maximum likelihood estimation of the labels. This is not justified anywhere, and there is no evidence to support the claim.\
> A: We empirically find that initializing DDIM with the predicted label mean results in slightly improved testing accuracy compared to the stochastic version. Since the initial Gaussian distribution is unimodal, and it serves as the sole source of randomness in the generation process, using the mode to obtain a deterministic and most probable prediction emerges as an intuitive choice.
>
> W3: The experiments on the WebVision and ILSVRC2012 datasets should be examined further. The method presented in this paper relies heavily on having a high-quality pre-trained encoder network that can provide meaningful representations during training. For these two experiments, the authors utilized CLIP, whereas the next best-performing approaches, C2D and NCR, were only trained using samples from the smaller datasets. It is possible that the majority of the performance increases shown in this paper can be attributed to the discriminative ability of the CLIP encoder, since additionally on the Clothing1M dataset, where CLIP is not expected to perform well, the authors did not manage to greatly outperform the previous state-of-the-art results.\
> A: As discussed in our global response, the KNN and linear probing results added to Table 3 illustrate that our method effectively incorporates CLIP features in supervised learning from noisy labels. Utilizing naive linear probing with neighboring labels degrades the pre-trained feature space, leading to diminished performance in comparison to the unsupervised KNN approach. Moreover, our method has the flexibility to incorporate other techniques to further enhance accuracy.
>
> Q1: For a single sample, is the frequency of the neighbors seen in training the same regardless of their distances in the latent space? If so, how is a neighborhood defined?\
> A: Neighbor is defined the same as in the K-nearest neighbor algorithm, which is by the rank of distances. We choose K based on KNN performance on the validation set as discussed in supplementary C2.

---

> > ### Author Response · Authors · 2023-08-20
> >
> > We wanted to follow up on the question you raised during the rebuttal phase. We hope that our responses have sufficiently addressed your concern. Should you have any further questions or require additional clarification, please do not hesitate to reach out. We stand ready to provide any additional explanations as needed.

---

### Official Review · Reviewer_KNdw · 2023-07-09

**Soundness:** 3 good
**Presentation:** 3 good
**Contribution:** 3 good
**Rating:** 7
**Confidence:** 4

**Summary:**

This paper considers image classification with noisy labels. Rather than consider a single label for each image, it considers the label distribution within the set of neighboring examples (determined using a pre-trained feature extractor), modelling this distribution by a learned diffusion process that is conditioned on image features, similar to CARD. The diffusion process maps a random label in the neighboring set to a normal distribution whose mean is either zero or the prediction of a baseline classifier, such that the reverse process should sample labels from the neighboring set. Inference is performed using DDIM to find the maximum likelihood label. The method assumes a pre-trained feature extractor (SimCLR trained on training data or CLIP trained on external data) is available for the purpose of finding neighbors and conditioning the diffusion process. The method is compared to strong baselines on several synthetic and real-world datasets and often provides a large improvement. Ablative studies demonstrate the importance of each component.

**Strengths:**

1. The ablative study (Table 2; CIFAR10/100 with synthetic noise) demonstrates that the combination of CLIP/SimCLR features and LRA-diffusion is key to the effectiveness of the method: linear probing, LRA (label retrieval) and CARD (diffusion-based classification) alone do not achieve such an improvement.
1. Idea is well motivated.
1. Good coverage of related work.
1. Well written.
1. Synthetic experiments use strong type of noise based on second-highest confidence.
1. Inference time included in empirical results.

**Weaknesses:**

1. Real-world evaluation (WebVision, ILSVRC, Food-101N) does not include simple baselines using CLIP features (KNN classifier and linear probe). This is important because it lets us verify that the results do not simply reflect the strength of the CLIP features.
1. Real-world evaluation does not include SimCLR features. This is important because it shows the performance of the method without external data. In general, it would be good to highlight which methods use (which) external datasets and/or pre-trained weights in the SOTA comparison.
1. (Sections 3.3 and 3.4) The modification of the DDIM procedure to allow non-zero mean seems to be redundant, since all experiments then used a zero-mean distribution by setting $f_q = 0$ (line 179)?
1. Unclear what "Linear probing + LRA" is in Table 2. Does this entail training a linear model using the average of the neighboring labels as the softmax-cross-entropy target?
1. It does seem inelegant to use a normal distribution to represent probability vectors (as acknowledged in text). Did the authors try working in real-valued logits instead?

Suggestions:
1. I'm not a big fan of naming the method "retrieval", since the term suggests to me that training data is retrieved from a larger, external source of training data (not the case).
1. Discussion of impact of $k$ (number of neighbors) is deferred to the supp mat and I didn't see any references to this in the main text. It would be good to at least reference it and ideally add a plot.
1. Could be good to include some discussion of singly-labelled vs multiply-labelled noisy labels, since the proposed method effectively constructs a multiply-labelled dataset from a singly-labelled dataset.

**Questions:**

Please address weaknesses.

**Limitations:**

The authors have identified limitations including:
1. dependence on pre-trained feature extractors
1. method is less effective beyond ~50% corrupted labels
1. normal distribution not ideal for working on probability simplex

I do not see any ethical issues.

---

> ### Author Rebuttal · Authors · 2023-08-10
>
> Q1: Real-world evaluation (WebVision, ILSVRC, Food-101N) does not include simple baselines using CLIP features (KNN classifier and linear probe). This is important because it lets us verify that the results do not simply reflect the strength of the CLIP features.\
> A: We add the KNN and Linear Probing results to Table 3 in the attached PDF. Similar to our ablation study, linear probing results in decreased performance compared to KNN. On the other hand, diffusion methods are effective and robust in incorporating CLIP features. The test accuracy on ILSVRC2012 shows a performance drop due to supervised training bias on Webvision. This limitation does not apply to the unsupervised KNN.
>
> Q2: Real-world evaluation does not include SimCLR features. This is important because it shows the performance of the method without external data. In general, it would be good to highlight which methods use (which) external datasets and/or pre-trained weights in the SOTA comparison.\
> A: We found that using SimCLR features is less effective compared to SOTA methods, partly due to the sub-optimality of the trained features. However, SimCLR+Diffusion (our method) still outperforms the simple SimCLR method. We also have shown improved results with CC, which isn't trained with external data, on the Clothing1M dataset, as presented in Table 5. This demonstrates that our method can achieve improved results when coupled with other feature-learning methods.
>
> Q3: (Sections 3.3 and 3.4) The modification of the DDIM procedure to allow non-zero mean seems to be redundant, since all experiments then used a zero-mean distribution by setting  (line 179)?\
> A: We would like to argue that we show that allowing DDIM with non-zero mean is a more theoretically general framework of our method, but do not advocate the exact adoption of the method. In practice, we do need to adopt the most practically effective setting. We empirically found that utilizing a noisy classifier as the $f_q$ encoder does not consistently lead to improved results. For simplicity, we thus only use a zero mean to reduce the model complexity.
>
> Q4: Unclear what "Linear probing + LRA" is in Table 2. Does this entail training a linear model using the average of the neighboring labels as the softmax-cross-entropy target?\
> A: An sample label is a one-hot label vector, sampled from neighboring labels as outlined in Algorithm 1, step 4, on page 4. We've included the result of the linear probing with mean label setting that you requested in Table 2 of the attached file. It is more effective than using sample labels for linear probing. Additionally, we have discussed the utilization of mean labels for diffusion in Supplementary D2, where we find that the performance is comparable to that of sample labels.
>
> Q5: It does seem inelegant to use a normal distribution to represent probability vectors (as acknowledged in text). Did the authors try working in real-valued logits instead?\
> A: We agree that the generated label vector is not confined to the probability simplex, but it can be interpreted as the logit of the prediction head of a classifier. To further address your concerns, we additionally performed experiments using diffusion on simplex following the current Star-shaped DDPM (Okhotin 2023). As stated in their paper, the utilization of Dirichlet distribution in a diffusion model requires a different mechanism design than DDPM. We have added the results as Dirichlet diffusion in Table 2. Although it has a one-hot label interpretation, the performance is indeed slightly worse than using standard DDPM. We hope this response adequately addresses your concern.
>
> * Okhotin, Andrey, et al. "Star-Shaped Denoising Diffusion Probabilistic Models." arXiv preprint arXiv:2302.05259 (2023).
>
> Suggestions:
>
> S1: I'm not a big fan of naming the method "retrieval", since the term suggests to me that training data is retrieved from a larger, external source of training data (not the case).\
> A: We use the term 'retrieval' in the name because the label is augmented by the neighbor's label, and the neighbors are retrieved from an external feature space, which can be pre-trained by external data. However, we understand how the term 'retrieval' might cause misunderstandings. Accordingly, we will consider changing the name to “label-corrected diffusion” to better reflect the actual function.
>
> S2: Discussion of impact of  (number of neighbors) is deferred to the supp mat and I didn't see any references to this in the main text. It would be good to at least reference it and ideally add a plot.\
> A: Thank you for your suggestion. We will add a reference to the impact of the number of neighbors in the main text and include a small plot in the supplementary material to address this concern.
>
> S3: Could be good to include some discussion of singly-labeled vs multiply-labeled noisy labels, since the proposed method effectively constructs a multiply-labeled dataset from a singly-labeled dataset.\
> A: Thank you for your suggestion. We plan to include a discussion of applying our method to multi-label datasets in the conclusion section, which we hope to explore as interesting future work.

---

> > ### Comment · Reviewer_KNdw · 2023-08-21
> >
> > > Q1 (baselines with CLIP features for real-world datasets), Q2 (baselines with SimCLR features for real-world datasets), Q4 (is mean label used for "linear probe + LRA"?)
> >
> > Thank you for conducting these experiments, this improves my confidence in the effectiveness of the method.
> >
> > > Q3 (framework is more general than experiments)
> >
> > This is ok but the reader should be warned that experiments adopt $f_q = 0$ at the time that it is introduced, with the justification that the authors have provided in the rebuttal.
> >
> > > Q5 (Gaussian not on simplex)
> >
> > Thanks for conducting this initial experiment. This is interesting and important for completeness.
> >
> > > S1 (proposal to use name "label-corrected diffusion")
> >
> > To me, "label-corrected" could mean anything. Maybe something like "neighborhood label distribution diffusion", but it's up to you!
> >
> > > S2 (number of neighbors), S3 (discussion of multiply-labelled noisy labels)
> >
> > Thanks for taking these suggestions on board.
> >
> > **Overall**
> >
> > The paper proposes a clever use of diffusion to incorporate neighborhood consistency in noisy-label learning. While neither component is novel, I find the application to be highly apt, and moreover the method is effective. The additional experiments in the rebuttal help demonstrate that the improvement is not simply due to the use of pre-trained or self-supervised features. I would prefer to see a greater discussion of other methods for predicting a multimodal distribution in the final version. I preserve my initial positive rating.

---

### Author Rebuttal · Authors · 2023-08-10

We thank the reviewers for their valuable comments. We are happy that the reviewers find our manuscript well-written, and demonstrate effective application of our method. A common issue raised by the reviewers is the incorporation of the CLIP feature which uses external data to train. We will address this issue below. For other specific questions raised by each reviewer, we will post our responses separately. We also have added requested experiment results in the reviews in the attached one page pdf file. We will incorporate more detailed revisions into the camera-ready version according to our responses to the reviews.

Q: The utilization of CLIP in the performance evaluation raises concerns because it involves the incorporation of additional data. It is important to conduct a fair comparison by addressing this aspect.\
A: We thank the reviewers for this general comment. We argue that the results do not simply reflect the strength of the CLIP features, for the following reasons. We will incorporate these arguments into the final revision for clarification.

1. We have added multiple baselines in the ablation study, the results are shown in Table 2 of the attached PDF. All the baseline methods use the pre-trained SimCLR and CLIP features. The SimCLR feature is trained on the same training data through contrastive learning without access to external dataset. The results demonstrate that our method is the key to the successful application of pre-trained features in learning from noisy labels. In addition, we want to emphasize that we are the first to develop a method that effectively uses the CLIP feature in learning with noisy labels. This innovation is a significant part of our contribution.

2. We show results of linear probing and KNN using CLIP features in Table 3 of the attached PDF to further demonstrate the effectiveness of our design. The results suggested that utilizing linear probing with LRA and mean labels degrades the pre-trained feature space, leading to diminished performance in comparison to the unsupervised KNN approach. On the other hand, the diffusion process can effectively incorporate the feature space to achieve higher performance.
In addition, we included results of linear probing using average of neighboring labels (denoted as mean) instead of sampling from neighboring labels (denoted as sample) in Table 2 in response to reviewer KNdw’s inquiry. We found that mean labels are more robust than sample labels for linear probing. However, we show both sample and mean label works with the diffusion model and result in similar results. We also included the diffusion result with mean labels to show it is slightly worse than sample labels in most cases on real-world dataset. We have also discussed the use of mean label in Supplementary D2.

3. In Table 5 of the manuscript, we have shown improved results with CC, which isn't trained with external data, on the Clothing1M dataset. This demonstrates that our method can achieve improved results when coupled with other feature-learning methods. In fact, we would like to emphasize that a common practice adopted by most existing methods to achieve state of the arts is by using pretrained models (why wouldn’t if the pretrained models are available?). In our setting, for example, NCR achieves SOTA by the combination of NCR+Mixup+DA, C2D also has to collaborate with ELR+, DivideMix, and SimCLR to achieve SOTA results. We will highlight these hybrid methods in our comparison. Since the LRA diffusion method is orthogonal to many existing techniques, we believe that by combining it with other learning techniques, such as Mixup augmentation and Co-teaching, our method will facilitate the development of new methods for learning from noisy labels in the future.

---

> ### Author Response · Authors · 2023-08-19
>
> Regarding the common concern about the integration of the CLIP feature, we wish to offer further insight and clarification:
>
> The integration of pre-trained models for learning from noisy labels is actively being explored due to the availability of the CLIP encoder and its potential to significantly enhance performance. Recently, a new method, EPL (Jongwoo Ko, 2023), has proposed to incorporate the most powerful CLIP and ConvNext-XL encoders and cooperate with other SOTA methods such as ELR+, DivideMix, and UNICON. However, our diffusion-based method outperforms EPL by achieving ~6% higher accuracy on WebVision and ILSVRC12 datasets. This improvement over EPL demonstrates that developing better ways to incorporate pre-trained models to facilitate learning from noisy labels is a non-trivial task and represents a valuable contribution of our method. We will include this comparison in our final version. We hope this clarification addresses your concern.
>
> * Ko, Jongwoo, Sumyeong Ahn, and Se-Young Yun. "EFFICIENT UTILIZATION OF PRE-TRAINED MODEL FOR LEARNING WITH NOISY LABELS." ICLR 2023 Workshop on Pitfalls of limited data and computation for Trustworthy ML. 2023.

---

### Decision · Program_Chairs · 2023-09-21

**Decision:**

Accept (poster)

**Comment:**

This paper takes a (deep) generative modeling approach to learning from noisy labels. The proposed approach leverages a diffusion model which iteratively refines a noisy label from an initial random guess. It is based on the recently proposed Classification and Regression Diffusion Models (CARD).

The paper largely received mostly positive reviews. However, one of the reviewers (Reviewer 21mt) expressed some serious concerns regarding novelty. The reviewer felt that the paper is a simple combination of classification diffusion model with retrieval augmentation. The reviewer also expressed concern about the fact that the experiments use a pre-trained CLIP encoder for feature extraction. The latter concern was shared by some of the other reviewers as well.

The authors submitted a detailed rebuttal arguing about the novelty aspect and also reported some additional ablation results showing that it is not just the CLIP encoder but the other aspects of the proposed method that contribute to the performance gain. One of the other reviewers also acknowledged this post-rebuttal.

Regarding the novelty aspects, my own assessment (which is shared by some of the other reviewers as well) is that even though the proposed method is a combination of existing ideas (classification diffusion model and retrieval augmentation), the paper is applying them in an important setting (learning from noisy labels) and that the method works well in practice.

In view of the above points, looking at the reviewers, the rebuttal, and the discussion, and my own reading of the paper, I recommend the paper for acceptance. The authors should incorporate the reviewers' suggestions and address the other points of criticism when preparing the final version.